# Brown adipose tissue CoQ deficiency activates the integrated stress response and FGF21-dependent mitohormesis

Ching-Fang Chang [ID][1,5], Amanda L Gunawan [ID][1,5], Irene Liparulo [ID][1], Peter-James H Zushin[1], Kaitlyn Vitangcol[1], Greg A Timblin[2], Kaoru Saijo[3], Biao Wang[4], Güneş Parlakgül [ID][1], Ana Paula Arruda[1] & Andreas Stahl [ID][1✉]

## Abstract

Coenzyme Q (CoQ) is essential for mitochondrial respiration and required for thermogenic activity in brown adipose tissues (BAT). CoQ deficiency leads to a wide range of pathological manifestations, but mechanistic consequences of CoQ deficiency in specific tissues, such as BAT, remain poorly understood. Here, we show that pharmacological or genetic CoQ deficiency in BAT leads to stress signals causing accumulation of cytosolic mitochondrial RNAs and activation of the eIF2α kinase PKR, resulting in activation of the integrated stress response (ISR) with suppression of UCP1 but induction of FGF21 expression. Strikingly, despite diminished UCP1 levels, BAT CoQ deficiency displays increased whole-body metabolic rates at room temperature and thermoneutrality resulting in decreased weight gain on high-fat diets (HFD). In line with enhanced metabolic rates, BAT and inguinal white adipose tissue (iWAT) interorgan crosstalk caused increased browning of iWAT in BAT-specific CoQ deficient animals. This mitohormesis-like effect depends on the ATF4-FGF21 axis and BAT-secreted FGF21, revealing an unexpected role for CoQ in the modulation of whole-body energy expenditure with wide-ranging implications for primary and secondary CoQ deficiencies.

**Keywords** Coenzyme Q; Brown Adipose Tissue; Mitochondrial Unfolded Protein Response; FGF21; Mitohormesis
**Subject Category** Metabolism

## Introduction

Coenzyme Q (CoQ) is a component of oxidative phosphorylation (OXPHOS) in the mitochondria and is responsible for carrying electrons from multiple entry points to Complex III. It is also the only known endogenously synthesized lipid-soluble antioxidant, protecting cellular lipids, proteins and DNA from oxidative events (Littarru and Tiano, 2010). While the functions of CoQ in the mitochondrial electron transport chain are well explored, a plethora of basic questions about CoQ physiology such as its transport (Anderson et al, 2015; Deshwal et al, 2023), inter-organellar distribution, participation in redox reactions outside of mitochondria (Bersuker et al, 2019), and roles in mitochondrial to nuclear communications are still not fully answered (Guerra and Pagliarini, 2023).

Given the remarkable biochemical properties of CoQ and its ubiquitarian distribution throughout the cell, CoQ deficiency has been associated with multiple human pathologies (Stefely and Pagliarini, 2017). Primary CoQ deficiencies result from mutations in genes involved in CoQ10 biosynthesis. Secondary CoQ deficiency has been frequently detected in mitochondrial disorders, such as patients with mitochondrial DNA depletion syndromes (Montero et al, 2013) and also can be linked to hydroxymethylglutaryl coenzyme A (HMG-CoA) reductase inhibitors, such as statins, which are commonly used in the treatment of hypercholesterolemia (Hargreaves, 2003). In fact, CoQ deficiency commonly affects organs rich in mitochondria and with high energy demand (Stefely and Pagliarini, 2017) and our past studies have revealed brown adipose tissue (BAT) as a mitochondria-rich tissue that is affected by CoQ deficiency.

Despite this, CoQ's multifaceted roles in mammalian metabolism regulation and its engagement in highly metabolic tissue, such as BAT, remain elusive. BAT exhibits a multilocular lipid droplet morphology and densely packed mitochondria (Enerback, 2010; Rosen and Spiegelman, 2014). Furthermore, BAT has increasingly drawn much attention due to its high metabolic activity and its ability to perform thermogenesis through the actions of uncoupling

[1]Department of Nutritional Sciences and Toxicology, University of California, Berkeley, Berkeley, CA 94720, USA. [2]Center for Bioengineering and Tissue Regeneration, Department of Surgery, University of California, San Francisco, San Francisco, CA, USA. [3]Department of Molecular and Cell Biology, University of California at Berkeley, Berkeley, CA 94720, USA. [4]Cardiovascular Research Institute, Department of Physiology, University of California, San Francisco, CA 94158, USA. [5]These authors contributed equally: Ching-Fang Chang, Amanda L Gunawan. ✉E-mail: astahl@berkeley.edu

protein 1 (UCP1), which is able to dissipate energy in the form of heat independent of ATP production. BAT has also been recognized as a secretory organ (Cypess, 2022). More than a decade after the discovery of BAT as a functional and secretory tissue in adult humans, there is emerging evidence suggesting BAT as a promising therapeutic target for treating various metabolic diseases and its comorbidities (Cypess et al, 2009; Cypess and Kahn, 2010). Therefore, tremendous efforts are carried out to elucidate mechanisms regulating BAT mitochondrial activity and its secretory function.

As approximately 99% of mitochondrial proteins are encoded in the nucleus, coordination of mitochondrial function and nuclear gene transcription are critical for cellular homeostasis (Quiros et al, 2017). The mitochondrial unfolded protein response (UPR$^{mt}$) can be triggered by different events such as improper mitochondrial protein import capacity (Nargund et al, 2012), accumulation of unfolded proteins (Rolland et al, 2019; Pimenta De Castro et al, 2012), inhibition of mitochondrial chaperones or proteases (Rolland et al, 2019), and the increase of reactive oxygen species (ROS) levels, as well as OxPhos impairment (Qureshi et al, 2017). This evolutionarily conserved response, has been extensively studied in yeast and C.elegans (Pimenta De Castro et al, 2012; Bennett and Kaeberlein, 2014), whereas components of ISR in mammals are not yet fully dissected (Tran and Van Aken, 2020). ISR has been evaluated as an essential element to UPR$^{mt}$ activation via selective translation of transcription factors, such as ATF4, ATF5 and DNA-damage-inducible transcript 3 (CHOP), and attenuation of global protein synthesis by activation of eukaryotic translation initiation factor 2 (eIF2) complex subunit 2α (eIF2α; Anderson and Haynes, 2020; Forsstrom et al, 2019; Ost et al, 2020). Previous studies intriguingly show that ISR plays a more important role in mammals than in *C. elegans* (Anderson and Haynes, 2020) in response to mitochondrial dysfunction.

Although UPR$^{mt}$'s roles and implications in different diseases are still unclear, activation of adaptive responses to mitochondrial stress can induce metabolic benefits (Yi et al, 2018) and promote longevity (Merkwirth et al, 2016; Nargund et al, 2012), in a process termed mitohormesis (Yun and Finkel, 2014). Interestingly, perturbing mitochondrial functions in the central nervous system, intestine, or muscle cells has been shown to trigger a systemic hormetic response and increase lifespan in *C. elegans* and *D. melanogaster* (Bar-Ziv et al, 2020; Durieux et al, 2011; Owusu-Ansah et al, 2013). Key players in this mitochondrial propagation signaling, are mitokines such as FGF21, which can impact distant tissues and cause a global metabolic response (Salminen et al, 2017; Tezze et al, 2019). Emerging evidence suggests that the UPR$^{mt}$/ISR plays a crucial role in allowing cells to adapt and survive in the face of mitochondrial dysfunction (Kim and Sieburth, 2020).

We previously showed that BAT CoQ uptake hinges on the scavenger receptor CD36, revealing that the absence of CD36 leads to cold-intolerance (Anderson et al, 2015), emphasizing the significance of CoQ in BAT and thermogenesis. We also uncovered an unexpected role for CoQ in the transcriptional regulation of pivotal thermogenic genes like UCP1, affecting uncoupled respiration and leading to a rise in the inner mitochondrial membrane potential and a decline in ADP/ATP ratios (Chang et al, 2022). These insights prompted us to delve deeper into the effects of CoQ deficiency in BAT. In this study, we report that a BAT-specific upregulation of ISR, induced by CoQ deficiency, unexpectedly resulted in an FGF21-dependent increase in whole-body metabolism. This occurred despite the downregulation of UCP1 expression and resulted in decreased susceptibility to diet-induced obesity. Further, we identified an interplay between BAT and peripheral tissues, such as iWAT, during BAT dysfunction. This interaction culminated in a mitohormetic effect, offering metabolic benefits.

## BAT CoQ deficiency activates ISR and UPR$^{mt}$

We previously found that CoQ deficiency in brown adipose tissue resulted in lower oxygen consumption rates due to suppression of uncoupling protein 1 (UCP1) expression (Chang et al, 2022). To further define the transcriptional signature of CoQ deficiency, we performed RNA sequencing (RNAseq) analysis of mature brown adipocytes treated with 4-chlorobenzoic acid (4CBA), a polyprenyl-diphosphate:4-HB transferase (COQ2) inhibitor to block CoQ biosynthesis, or vehicle for 24 h. Analysis of the predicted subcellular locations of the top 50 upregulated and top 50 downregulated genes revealed that 4CBA treatment affected proteins in multiple organelles. Genes encoding proteins with multiple subcellular organelle locations are colored blue. There was a noticeable bias toward genes encoding secreted proteins (31 of 100 genes), followed by genes encoding cytosolic proteins (23 of 100 genes), and genes encoding plasma membrane-localized proteins (20 of 100 genes). 19 of the 100 top genes encoded proteins localized in the nucleus, showing a potential change in transcriptional regulation during BAT CoQ deficiency. Genes encoding mitochondrially localized proteins accounted for 16 of the top 100 genes (Fig. 1A). Since CoQ plays a vital role in the mitochondria, separate analysis of the top 50 upregulated and top 50 downregulated mitochondrial genes was done. These included genes for proteins involved in essential mitochondrial functions such as respiration, mitochondrial organization, and dicarboxylic acid metabolism (Figure EV1A). Genes in HIF1α and Nrf2 pathways, known mitochondrial ROS (mtROS) targets (Hamanaka and Chandel, 2009; Kasai et al, 2020), were induced by 4CBA treatments (Figure EV1B). Interestingly, cytosolic ROS scavenging enzymes and GSH synthesis were upregulated while mitochondrial antioxidants were suppressed (Figure EV1C), suggesting a unique redox alteration by CoQ deficiency. Gene Ontology (GO) analysis identified enrichment in mitochondrial dysfunction for categories including TCA cycles, respiratory electron transport and lipid catabolic processes repressed by 4CBA (Figure EV1D). Key components identified in endoplasmic reticulum and mitochondrial unfolded protein responses (UPR$^{mt}$) were increased by 4CBA (Figs. 1B and EV1E). Differential expression analysis confirmed UCP1 to be among the genes significantly reduced in expression. Up-regulated genes included notable ISR components such as ATF4, ATF5, CHOP, and FGF21 (Fig. 1C). HOMER motif analysis also revealed CHOP, ATF4 and C/EBP binding motif, key UPR$^{mt}$ and ISR effectors (Huggins et al, 2015; Pakos-Zebrucka et al, 2016), as top enriched transcription factor binding motifs in the promoters of the significantly upregulated genes (Fig. 1D). To confirm the RNAseq results, protein expression of UCP1, FGF21, ATF3 and ATF4 were measured after 48 h of 4CBA treatment. UCP1 expression was robustly downregulated by more than 70% while FGF21, ATF3 and ATF4 expression were all increased following 4CBA treatment (Fig. 1E). To examine BAT CoQ deficiency in vivo, we used a brown/beige fat-specific knockout of

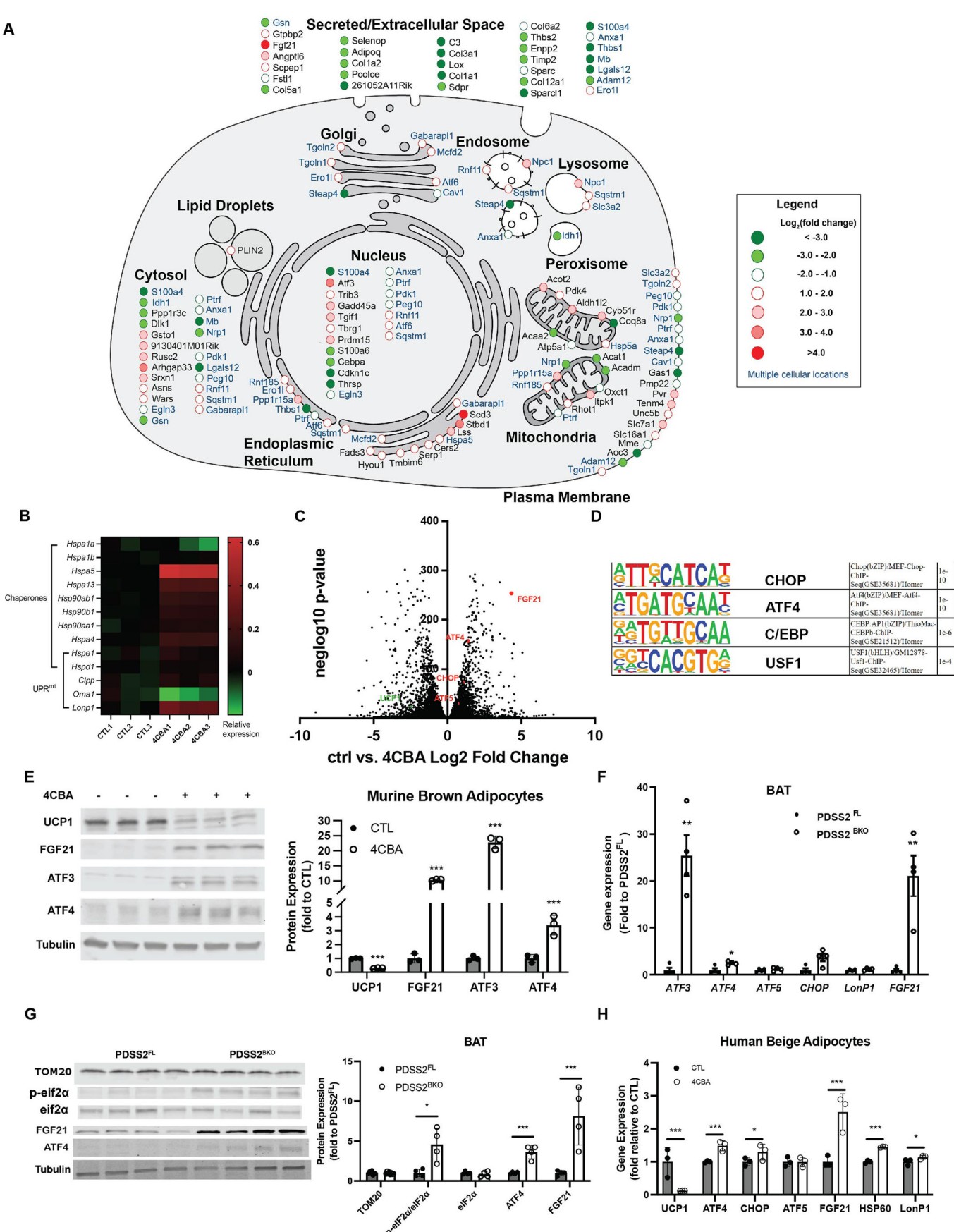

◄ **Figure 1.  BAT CoQ deficiency activates ISR and UPR$^{mt}$.**

(**A**) Cell map of RNA sequencing results from 4CBA treated cells. Includes overall top 50 upregulated and top 50 downregulated genes with p-value ≤ 5.91E-192. Genes encoding proteins with multiple subcellular locations are highlighted in blue. (**B**) Relative expression (log$_{10}$ -transformed RPKM value) of UPR-associated genes in murine brown adipocytes treated with 4CBA (24 h). (**C**) Volcano plot obtained from DESeq2 analysis of vehicle control and 4CBA-treated murine brown adipocytes RNA pools, $n = 3$ independent experiments/treatment. The Wald test was used for statistical analysis. (**D**) The top 3 enriched transcription factors identified by Motif analysis. (**E**) ISR-related protein levels in brown adipocytes after 4CBA treatment. (**F, G**) Gene or protein expression in brown adipose tissue (BAT) of PDSS2 floxed (PDSS2$^{FL}$) or BAT-specific PDSS2 knockout (PDSS2$^{BKO}$) animals, $n = 4$. (**H**) Gene expression analysis of human beige adipocytes treated with 4CBA for 48 h, $n = 3$. Data information: (**E–H**) Data are mean ± SEM. Results were compared using an unpaired two-tailed Student's $t$ test. Significance presented at *$P < 0.05$, **$P < 0.01$, and ***$P < 0.001$ compared to controls. Data describes biological replicates. Source data are available online for this figure.

the CoQ synthesis enzyme decaprenyl diphosphate synthase subunit 2 (PDSS2) by crossing UCP1-cre animals with a PDSS2 Flox (Peng et al, 2008) strain, both in the C57 background, generating the PDSS2$^{BKO}$ line. This model has previously been shown to have a 75% reduction in brown adipose tissue (BAT) CoQ levels, compared to floxed controls (PDSS2$^{FL}$), while other tissue types are unaffected, resulting in UCP1 suppression in BAT (Chang et al, 2022). The knockout was verified by measuring that PDSS2 gene expression was decreased in BAT, but not other organs such as liver (Figure EV1F). Similar to 4CBA treated cells, BAT from the PDSS2$^{BKO}$ mice have a pronounced upregulation of genes involved in the ISR and UPR$^{mt}$ (Fig. 1F,G). Stress response activation was also observed in human beige adipocytes in response to 4CBA treatment (Fig. 1H).

In addition to the in vivo genetic model we present, we also established genetically induced CoQ deficiency in an immortalized murine brown adipocyte cell line via siRNA driven silencing of the CoQ biosynthetic enzyme COQ2, the target of 4CBA treatment. Silencing of COQ2 RNA was carried out once brown adipocytes were mature. This model was congruent with the observed phenotype from 4CBA treated cells. There was a significant decrease in COQ2 gene expression by 75% and CoQ levels by 30% in COQ2 knockdown cells (Figure EV2A,B). There was activation of the stress response as noted by a twofold increase in gene expression of FGF21 and an increase in protein expression of ATF4 (Figure EV2B,D). A 60% decrease in UCP1 gene and 40% decrease in protein expression (Figure EV2B,C) was also observed.

## CoQ deficiency affects mitochondrial morphology and respiration in brown adipocytes

Next, we wanted to understand how mitochondrial structural dynamics are affected by BAT CoQ deficiency. Previous studies have shown that mitochondrial fission is required for thermogenic activities of brown and beige adipocytes (Pisani et al, 2018; Wikstrom et al, 2014). Indeed, congruent with the reduced UCP1 activity, mitochondria in 4CBA treated brown adipocytes are elongated and tubular differing from the fragmented mitochondria of the control group (Fig. 2A). They are also less round and have a higher aspect ratio, a key morphologic parameter indicating the ratio between the length of the major axis and the length of the minor axis (Fig. 2B). In accordance with these structural changes, phosphorylation of dynamin-related protein 1 (Drp1) at serine 616, required for facilitating mitochondrial fission (Wikstrom et al, 2014), was decreased by 25% in 4CBA treated cells (Fig. 2C). This was accompanied by a mild increase of mitochondrial mass as measured by VDAC expression (Fig. 2C). Transmission electron microscope (TEM) images were then used to gain further

understanding of these ultrastructural changes to mitochondrial morphology. Although brown adipocytes are known to have dense cristae (Frontini and Cinti, 2010), these images revealed that mitochondria in 4CBA treated cells have dramatically decreased cristae potentially resulting from impaired cristae formation and/or adaptive changes in response to stress and changes in metabolic needs (Fig. 2D). To quantify cristae density, the ratio of cristae length to mitochondria perimeter was measured and was significantly decreased in 4CBA treated cells (Fig. 2E).

In line with these changes to mitochondrial morphology and decreased UCP1 expression, COQ2 silencing experiments lowered oxygen consumption rates (OCR), regardless of the available substrate (Figure EV2E,G). Under glucose-rich conditions basal, maximal and proton leak respiration rates were decreased (Figure EV2F). Basal and maximal respiration was also decreased in COQ2 knockdown cells when palmitate was available as the main substrate, both in the presence or absence of etomoxir, an inhibitor of carnitine palmitoyl transferase-1, a crucial rate-limiting enzyme in fatty acid oxidation (Figure EV2H). Additionally, the mitochondrial morphology changes observed in 4CBA-treated brown adipocytes were mirrored in COQ2 knockdown cells (Figure EV2I).

## BAT CoQ deficiency triggers cytosolic accumulation of mitochondrial RNA and PKR activation

To better understand the mechanism by which the ISR is induced within CoQ deficient brown adipocytes, we sought to identify the involved molecular components. Previous published work on ISR has established a mechanism by which cellular stress is sensed by members of the eIF2α kinase: PERK, GCN2, HRI and PKR (Taniuchi et al, 2016), triggering their activation. These kinases then phosphorylate the α subunit of eIF2 at ser51 leading to inhibition of global translation and upregulation of translation of mRNAs containing multiple upstream open reading frames (uORFs) including ATF4 (Harding et al, 2000). Increased amounts of ATF4 protein, a transcription regulator, results in increased transcription of genes involved in the ISR (Harding et al, 2003; Fig. 3A). The core event in the ISR is the phosphorylation of eIF2α by stress kinases. To identify which stress kinase is implicated in CoQ-mediated ISR resulting in UCP1 suppression, we treated cells with 4CBA and pharmacological inhibitors of eIF2α kinases or other components of the ISR (Pakos-Zebrucka et al, 2016) pathway including PKR inhibitors imoxin and 7DG, GCN2 inhibitor A92, PERK inhibitor GSK2606414 (GSK) and ISRIB. While the GCN2 inhibitor A92 failed to rescue the UCP1 suppression phenotype, the PERK inhibitor GSK, and the reversible PKR inhibitors 7DG and ISRIB had mild, but significant effects, in 4CBA treated cells (Fig. 3B). As it has been suggested that HRI is required for ISR activation upon mitochondria stress (Guo et al, 2020), we used siRNA targeting HRI to examine the involvement of HRI

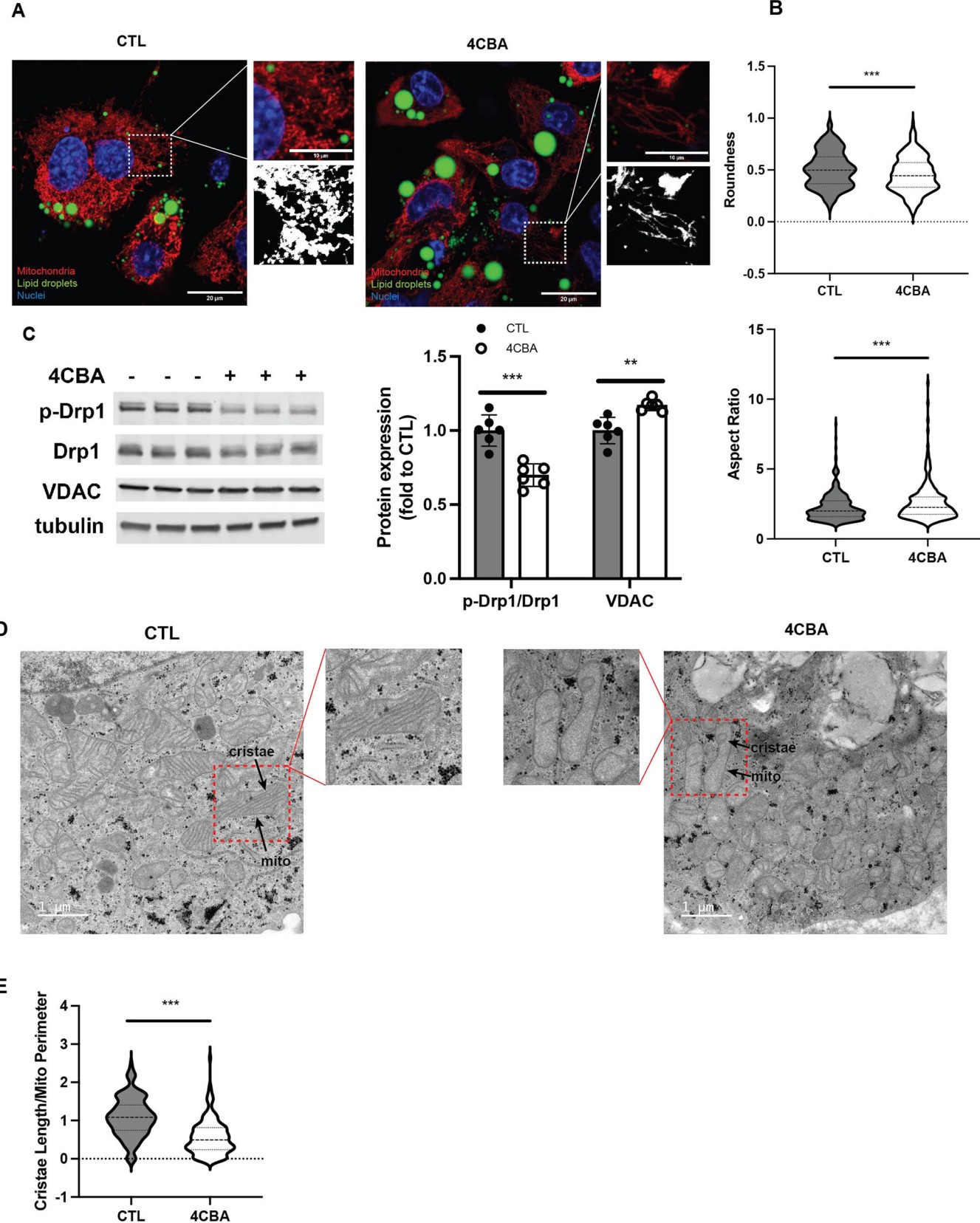

**Figure 2.  CoQ deficiency affects mitochondrial morphology in brown adipocytes.**

(**A**) Brown adipocyte with vehicle (CTL) or 4CBA treatment were stained with MitoTracker deep red, bodipy and dapi and visualized using the Zeiss LSM710, scale bar = 20 μm. (**B**) The roundness and aspect ratio of the visualized mitochondria were assessed using Fiji from 2 independent experiments. (**C**) Phosphorylation of Drp1 was assessed and quantified using western blot, $n = 6$. (**D**) Mitochondrial morphology of brown adipocytes with treatment was visualized using TEM with the Tecnai-12, scale bar = 1 μm. (**E**) Cristae Length/Mito Perimeter of visualized mitochondria were assessed using Fiji from 3 independent experiments. Data information: (**C**) Data are mean ± SEM. (**B, C, E**) Results were compared using an unpaired two-tailed Student's *t* test. Significance presented at **$P < 0.01$, and ***$P < 0.001$ compared to controls. Data describes biological replicates. Source data are available online for this figure.

in ISR dependent UCP1 suppression by CoQ deficiency (pharmacological inhibitors are unavailable). While siRNA treatment resulted in a robust 60% reduction of HRI transcripts, it did not rescue UCP1 expression significantly (Figure EV3A). Interestingly, treatment of CoQ deficient cells with the PKR inhibitor imoxin robustly rescued UCP1 gene expression (Fig. 3B) indicating a key role for this eIF2α kinase in the ISR activation by CoQ deficiency. The activation of PKR in murine brown adipocyte cells was further investigated by using a $Zn^{2+}$-Phos-tag acrylamide gel (Kinoshita and Kinoshita-Kikuta, 2011) to separate phosphorylated and unphosphorylated forms of PKR. Phosphorylated proteins are bound by $Zn^{2+}$ and are detected as bands with an upward mobility shift. PKR phosphorylation was significantly increased at both 7 and 24 h after 4CBA treatment, observed by an increase in signal of the uppermost PKR band, showing activation of PKR at these timepoints (Fig. 3C). This was further validated by studying the phosphorylation of the Thr451 residue on PKR subsequent to 4CBA treatment in murine brown adipocytes which was increased after 24 h of treatment (Figure EV3B). Phosphorylation status of PERK and GCN2 were also studied using $Zn^{2+}$-Phos-tag acrylamide gels. 2.5 μg/mL of tunicamycin, an activator of the unfolded protein response, was used as a positive control for PERK phosphorylation (Figure EV3C). PERK and GCN2 phosphorylation were not significantly increased with 4CBA treatment, allowing us to identify PKR as the main kinase participating in the integrated stress response triggered by CoQ deficiency. PKR can be activated by viral RNAs but also by cytoplasmic mitochondrial double-stranded RNA (mtRNAs; Kim et al, 2018; Sud et al, 2016). We found that levels of mitochondrial transcripts of ND1, ND4, ND5 and CYTB were increased in the cytoplasmic fraction, but decreased in the mitochondria of the 4CBA-treated brown adipocytes (Fig. 3D). This suggests that mitochondrial CoQ deficiency induces the ISR via increased mtRNAs export and PKR-mediated eIF2α phosphorylation. Successful fractionation of cells was verified by measuring SDHB expression, a nuclear-encoded mitochondrial protein, which was enriched in the cytosolic, but not the mitochondrial fraction (Figure EV3D).

## Induction of ISR precedes suppression of UCP1 expression

After identifying PKR as the stress kinase activated upon CoQ deficiency in BAT, we wanted to study the temporal effects of ISR activation. We first measured CoQ levels after 2, 7, 24, 48 and 72 h of 4CBA treatment. CoQ levels were already significantly decreased to 75% of controls after 2 h of 4CBA treatment, followed by a steady decrease in CoQ levels until 72 h when around 50% of cellular CoQ was depleted (Fig. 4A). Phosphorylation of eIF2α is the principal event leading to activation of ISR (Fig. 3A). Interestingly, phosphorylation of eIF2α was highest at 7 h of 4CBA treatment showing that even acute decreases in CoQ levels to 75% of control cells can trigger ISR activation. Phosphorylation of

eIF2α was significantly elevated until 48 h of 4CBA treatment (Fig. 4B). The next event downstream of eIF2α phosphorylation is increased translation of ATF4 (Fig. 3A), another early ISR regulator. Similar to eIF2α phosphorylation, ATF4 protein expression peaked at 7 h and drastically decreased, although still significantly increased compared to control cells, after 24 h of 4CBA treatment (Fig. 4C). ATF4 is known to act as a transcriptional regulator, increasing transcription of itself (Jin et al, 2009) as well as other genes involved in both ISR and UPR$^{mt}$ (Quiros et al, 2017). Indeed, induction in transcription of *ATF3, -4*, and *-5* was also rapid, peaking between 7 to 24 h, and stayed significantly elevated above baseline (Fig. 4D). The ISR related genes (Forsstrom et al, 2019) *FGF21, GDF15, MTHFD2*, and *PHGDH* also rose rapidly at 7 h and stayed robustly elevated throughout all treatment time points (Fig. 4E). UPR$^{mt}$ associated genes such as *HSP60, LONP1*, and *CLPP* were upregulated after ISR associated genes. UPR$^{mt}$ genes rose rapidly, peaking at 24 h and subsequently declined (Fig. 4F). Next, we wanted to understand how suppression of UCP1 relates to ISR activation. UCP1 gene expression was already significantly decreased to around 50% after 2 h of 4CBA treatment, showing a rapid response to CoQ deficiency, with a peak decrease in expression after 48 h where expression was less than 10% of control cells (Fig. 4G). UCP1 protein expression was not significantly decreased at 7 h and only 50% decreased at 24 h, suggesting that the suppression of UCP1 is transcriptionally regulated (Fig. 4H). While UCP1 suppression continues up until 72 h of 4CBA treatment, induction of ISR and UPR$^{mt}$ related genes peak in expression after 7 or 24 h, indicating a well-orchestrated cellular response to CoQ depletion whereby peak activation of ISR and UPR$^{mt}$ precedes peak suppression of UCP1.

To identify if ISR effectors are responsible for UCP1 suppression as a consequence of CoQ deficiency, we attempted to rescue UCP1 expression in CoQ-deficient brown adipocytes, via knockdown of ATF4, ATF5, and CHOP. SiRNA-mediated knockdown efficiency was comparable at 50% (Figure EV4A). While knockdown of ATF5 or CHOP alone could not significantly rescue UCP1 gene expression in 4CBA treated cells (Figure EV4B), ATF4 knockdown rescued UCP1 expression to 60% of cells treated with a scramble siRNA (SCR), which could be further increased by simultaneous CHOP, but not ATF5, knockdown (Fig. 4I), suggesting an involvement of an ATF4/CHOP heterodimer in UCP1 regulation in response to ISR.

## BAT CoQ deficiency modulates whole body metabolism

After observing induction of the ISR and UPR$^{mt}$, we wanted to further examine how the stress response triggered by BAT CoQ deficiency impacts whole body metabolism. Further, since FGF21 expression is increased, we wondered how this would affect whole body energy expenditure in vivo. We explored the metabolic

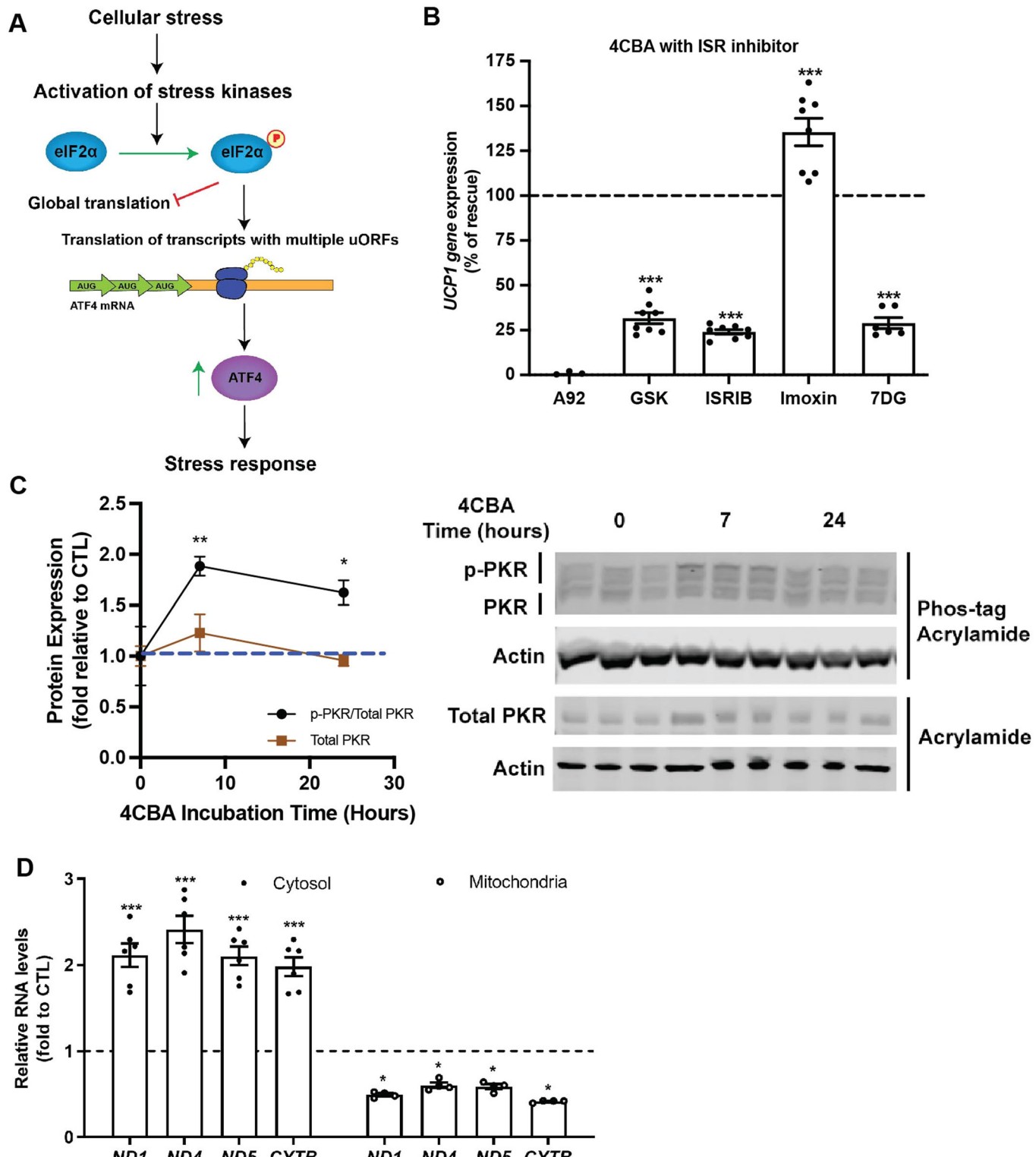

alterations on BAT-specific PDSS2 knockouts using CLAMS metabolic chambers at thermoneutrality (30 °C) and a mild cold challenge (23 °C). Mice were fed a low-fat defined diet for 2 to 3 weeks prior to being put into the CLAMS metabolic chambers to lower exogenous CoQ intake. Previously we had observed that, in

line with their low UCP1 expression in BAT, when PDSS2[BKO] mice were subjected to a 4 °C cold challenge, they were cold intolerant and exhibited dropping $VO_2$ rates (Chang et al, 2022). Due to this previous finding, we were surprised to observe that $VO_2$ rates during the light cycle (low activity) and dark cycle (high activity)

◄ **Figure 3.    BAT CoQ deficiency triggers cytosolic accumulation of mitochondrial RNA and PKR activation.**

(A) Diagram of known integrated stress response mechanism. Cellular stress can include ER stress, viral infection, amino acid starvation, oxidative stress, iron deprivation. Stress kinases include PERK, HRI, GCN2 and PKR. The stress response includes upregulation in expression of stress genes such as chaperones, FGF21, ATFs, GDF15, etc. (B) UCP1 gene expression in brown adipocytes treated with 4CBA in the presence of ISR inhibitors, $n = 6–8$. (C) Phosphorylated PKR was assessed using a phos-tag gel and compared to total PKR levels, $n = 3$. (D) Mitochondrial transcript levels in cytosol or mitochondria, $n = 4–6$. Data information: (B–D) Data are mean ± SEM. Results were compared using an unpaired two-tailed Student's $t$ test. Significance presented at $*P < 0.05$, $**P < 0.01$, and $***P < 0.001$ compared to controls. Data describes biological replicates. Source data are available online for this figure.

were increased in PDSS2[BKO] animals compared to PDSS2[FL] controls. This held true both at room temperature, a mild cold challenge for mice, and at thermoneutrality when BAT respiration should be suppressed (Fig. 5A–C) The increase in $VO_2$ rates also remained when data was normalized to body weight (Figure EV5A–C). Lower ambulatory activity was measured in the PDSS2[BKO] animals compared to PDSS2[FL] controls (Fig. 5D) indicating that the increased respiration is irrelevant to physical activity. A parallel metabolic study was performed to study if differences in food intake could explain the changes in respiration rates. No significant difference was observed in food intake among the two groups in the 3-week period preceding the metabolic cage experiment (Figure EV5D). Afterwards, real-time food consumption was monitored while mice were in the metabolic cages to examine if increased food consumption may explain the increases in metabolic rates within the PDSS2[BKO] animals (Figure EV5E). There were no significant food consumption differences at 23 °C and interestingly, at thermoneutrality PDSS2[BKO] had decreased food consumption compared to PDSS2[FL], suggesting that increases in metabolic rates were not due to increased food consumption in the knockout animals. Further, lean body mass composition was analyzed and was not significantly different between the two groups (Figure EV5F). This showed that although PDSS2[BKO] animals have lower ambulatory activity, this is not due to muscle wasting in the PDSS2[BKO] animals. To assess glucose metabolism in the PDSS2[BKO] mice, we performed a glucose tolerance and insulin tolerance test on these mice after 2 weeks of low-fat defined diet feeding. Despite the enhanced respiration we observed, there were no significant differences in glucose or insulin tolerance in PDSS2[BKO] mice (Figure EV5G–J). We then assessed whether PDSS2[BKO] animals were protected from high fat diet induced weight gain. In line with their enhanced respiration rates, PDSS2[BKO] gained significantly less body weight and had lower fat composition than control littermates when animals were fed with a 60% high fat diet for 12 weeks (Fig. 5E,F).

## ATF4-FGF21 axis is crucial in the metabolic adaptation to BAT CoQ-deficiency

In addition to its thermogenic function, BAT is also an endocrine tissue secreting several important metabolism regulating hormones including FGF21 (Villarroya et al, 2017). FGF21 is the master metabolic regulator in upregulation of energy expenditure and has promising therapeutic application for the treatment of obesity-related metabolic disorders (Geng et al, 2020). Thus, we wanted to explore how levels of this hormone change and affect metabolic rates in PDSS2[BKO] mice. Interestingly, not only did 4CBA treatment rapidly induced a 30 -to -40-fold increase in FGF21 transcripts in brown adipocytes in tissue culture (Fig. 4E) but an increase in FGF21 was also observed in BAT of PDSS2[BKO] animals (Fig. 6B).

Enhanced FGF21 expression by BAT also translated to 4–5-fold increase in circulating FGF21 levels (Fig. 6C). FGF21 expression remained unchanged in other tissues including the liver, which is considered the main site of FGF21 production (Badman et al, 2007), or inguinal white adipose tissue (iWAT). These data suggests that the observed elevation of FGF21 in circulation was mainly secreted by BAT (Fig. 6D,E) and the enhanced FGF21 secretion is part of the compensatory mechanisms downstream of BAT CoQ deficiency. Moreover, ISR markers were not elevated in the liver of liver-specific PDSS2 knockout animals (PDSS2[AlbKO]) (Appendix Fig. S1A), indicating the ISR induced by CoQ deficiency is tissue-specific.

To evaluate how enhanced FGF21 secretion in CoQ-deficient BAT influences whole-body metabolism in PDSS2[BKO] animals, we generated BAT-specific PDSS2 and FGF21 double knockouts (DKO[FGF21]) by crossing PDSS2[BKO] and FGF21 Flox animals. The tissue-specific knockout was verified via qPCR, showing that PDSS2 and FGF21 gene expression was decreased in the BAT of DKO[FGF21] animals, but not in the liver (Fig. 6A). These animals were fed a low-fat defined diet prior to analysis of their phenotype. BAT from DKO[FGF21] showed an increase in ISR-related genes ATF3, -4, -5 and CHOP, like those observed in BAT of PDSS2[BKO]. However, the induction of FGF21 expression in BAT of DKO[FGF21] animals was blunted (Fig. 6B) and was reflected in the circulating level of FGF21 (Fig. 6C). Decreased adiposity of the PDSS2[BKO] animals was normalized in the DKO[FGF21] animals, as there was no significant difference in body weight or fat composition as measured by EchoMRI (Fig. 6F,G). $VO_2$ rates during the light cycle and dark cycle were significantly decreased in DKO[FGF21] animals compared to PDSS2[FL] controls or PDSS2[BKO] animals in either room temperature (23 °C) or in thermoneutrality (30 °C) (Fig. 6H–J). These findings reveal that the whole-body metabolic alterations observed in the PDSS2[BKO] animals stem from increased BAT derived circulating FGF21. Enhanced expression of FGF21 has been linked to ISR pathway activation (Salminen et al, 2017). Since ATF4 is known to work upstream of FGF21 as an early ISR effector, we also studied the metabolic phenotype of BAT-specific dual ATF4/PDSS2 knockout animals (DKO[ATF4]) (Appendix Fig. S1B). DKO[ATF4] also abolished the increased expression of FGF21 and other stress response mediators in BAT as well as increased circulating FGF21 (Appendix Fig. S1C,D). Further, body weight, fat composition and $VO_2$ (Appendix Fig. S1E–I) were all also normalized in DKO[ATF4] animals, confirming the regulatory role of the ATF4-FGF21 axis on metabolic regulation by BAT CoQ deficiency.

## BAT CoQ deficiency induces mitohormetic responses in peripheral tissues

After observing FGF21 driven enhanced whole body respiration rates in PDSS2[BKO] mice, we wanted to understand if BAT or other

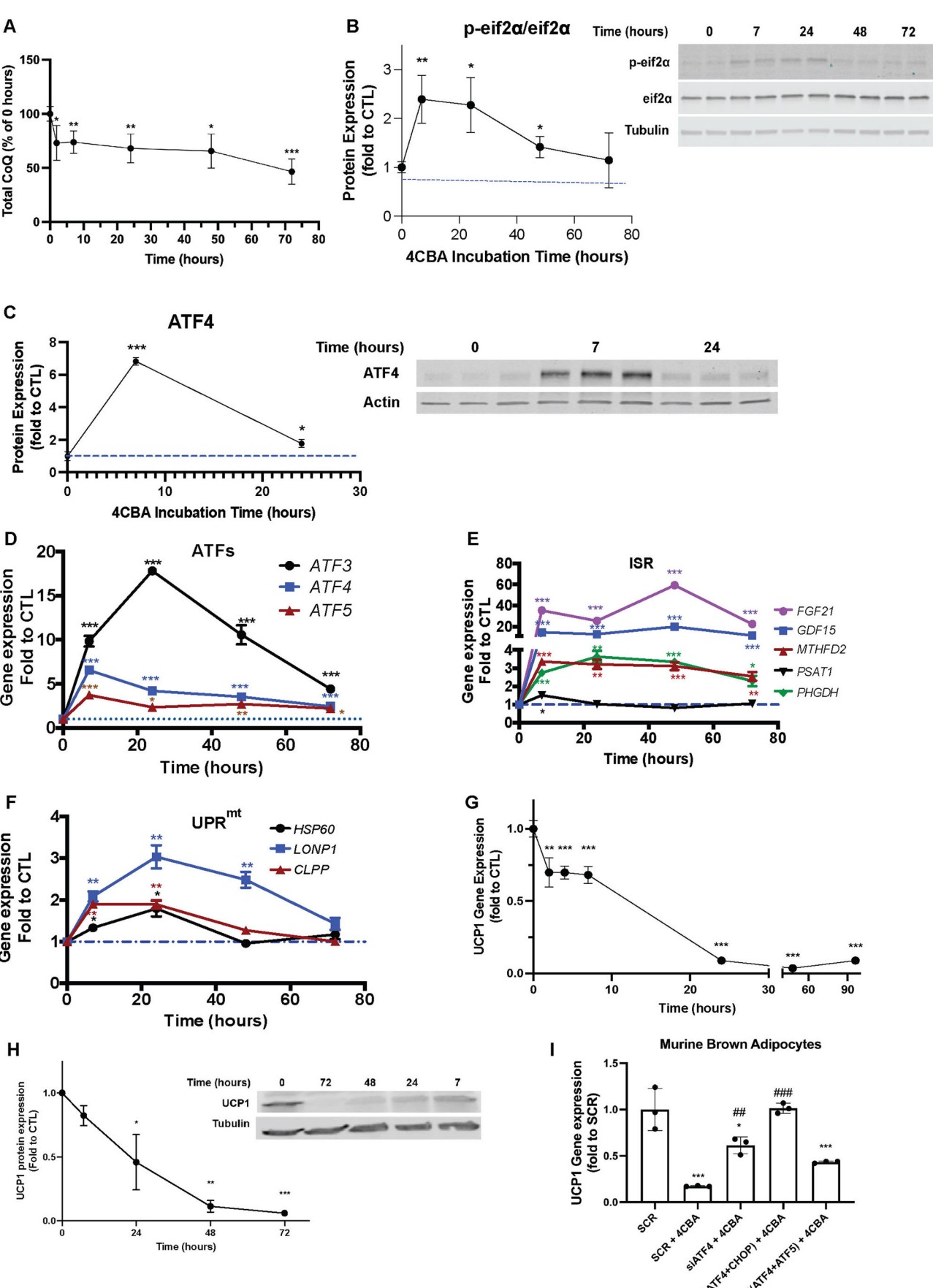

◄ **Figure 4. Induction of ISR precedes UCP1 expression suppression.**

(A) Total CoQ levels, including $CoQ_9$ and $CoQ_{10}$ isoforms, assessed and quantified after 0, 2, 7, 24, 48, and 72 h of 4CBA treatment. (B) Phosphorylated eIF2a protein levels were assessed and quantified after 0, 7, 24, 48, and 72 h of 4CBA treatment. (C) ATF4 protein levels were assessed and quantified after 0, 7 and 24 h of 4CBA treatment. (D–F) rtPCR analysis of genes in murine brown adipocytes treated with vehicle (CTL) or 4CBA for 0, 7, 24, 48, and 72 h. (G) UCP1 gene expression of brown adipocytes treated with 4CBA for 0, 2, 4, 7, 24, 48, and 96 h. (H) UCP1 protein expression of brown adipocytes treated with 4CBA for 0, 7, 24, 48, and 72 h. (I) UCP1 gene expression in brown adipocytes with siRNA knockdown of ATF4, ATF4 and CHOP or ATF4 and ATF5 following 24h-4CBA treatment. Results were compared using a one-way ANOVA test. Significance presented at ##$P < 0.01$, ###$P < 0.001$ compared to SCR + 4CBA. Data information: (A–I) Data are mean ± SEM with $n = 3$–5. (A–H) Results were compared using an unpaired two-tailed Student's $t$ test. Significance presented at *$P < 0.05$, **$P < 0.01$, and ***$P < 0.001$ compared to controls. Data describes biological replicates. Source data are available online for this figure.

organs were contributing to this surprising phenotype. To understand the BAT phenotype, oxygen consumption measurements were carried out in BAT tissue pieces isolated from PDSS2[FL] and PDSS2[BKO] mice. PDSS2[BKO] mice BAT tissue pieces showed a slight decrease in respiration compared to BAT tissue pieces from floxed controls (Appendix Fig. S2A,B). These findings from isolated BAT tissue are further supported by OCR measurements of immortalized murine brown adipocytes with COQ2 silencing (Figure EV2E,F), indicating that BAT is not the organ responsible for the enhanced respiration we observe in PDSS2[BKO] mice. Since iWAT and liver are both known mediators of FGF21 activity (Inagaki et al, 2007), oxygen consumption rates of iWAT tissue pieces and primary hepatocytes isolated from PDSS2[FL] and PDSS2[BKO] mice were measured next. iWAT tissue pieces from PDSS2[BKO] mice had significantly increased (1.4-fold) oxygen consumption rates compared to PDSS2[FL] iWAT. Unlike PDSS2[FL] iWAT, PDSS2[BKO] iWAT was unresponsive to FCCP treatment, suggesting that baseline respiration in PDSS2[BKO] iWAT is already at a maximum capacity for the tissue (Fig. 7A,B). Congruent with the increase in respiration rates, iWAT lobes from PDSS2[BKO] mice were noticeably smaller than lobes from PDSS2[FL] mice (Fig. 7D). In contrast, primary hepatocytes isolated from PDSS2[FL] and PDSS2[BKO] mice had no significant differences in oxygen consumption rate (Appendix Fig. S2E,F). We next wanted to identify what pathways in iWAT were leading to the observed increased respiration rates. Past publications have reported that FGF21 is known to increase thermogenic gene expression as well as beiging in iWAT (Fisher et al, 2012). Indeed, we found a threefold increase in gene expression of UCP1 in PDSS2[BKO] iWAT (Fig. 7E). However, this activation of the UCP1 promotor did not lead to strong induction of cre expression in iWAT as PDSS2 levels were only marginally decreased and iWAT CoQ levels remained unaffected (Fig. 7F). This suggests that the iWAT phenotype is a result of a mitohormetic BAT-derived signal and not secondary CoQ deficiency within iWAT. Immunohistochemistry analysis of iWAT revealed a striking beiging phenotype. UCP1 protein expression, pictured in red, was markedly increased in PDSS2[BKO] iWAT and localized around beige adipocytes with smaller, multilocular lipid droplet morphology, while PDSS2[FL] iWAT had little to no UCP1 expression and a unilocular lipid droplet morphology (Fig. 7G).

To determine other pathways in iWAT and peripheral tissues that are upregulated by FGF21 secretion, we studied changes in expression of alternative thermogenic genes. It is known that FGF21 can induce UCP1-independent thermogenic processes in adipose and other tissues (BonDurant et al, 2017; Chen et al, 2017). Interestingly, in addition to enhanced iWAT UCP1 expression and beiging, we found upregulation of genes in mechanisms of UCP1-independent thermogenesis in iWAT of PDSS2[BKO] animals. This

included *CPT1b* involved in fatty acid futile cycling (Guan et al, 2002) and mitochondrial ADP/ATP carrier *AAC2* suggested to promote AAC-mediated H+ leak (Bertholet et al, 2019; Fig. 7H). In BAT upregulated genes included *SLC6A8* and *GATM* in creatine-dependent thermogenesis (Kazak et al, 2015), *Ryr2* and *SERCA2* in calcium cycling (Ikeda et al, 2017; Ukropec et al, 2006), and *AAC2* (Appendix Fig. S2D). In iWAT, the activation of UCP1-independent thermogenesis was abolished in DKO[FGF21] and DKO[ATF4] knockout animals, while in BAT upregulation of the pathways remained in DKO[FGF21] animals but were abolished in DKO[ATF4] animals. This emphasizes the regulatory roles of ATF4 in BAT and the endocrine action of FGF21 in affecting the alternative thermogenesis in peripheral tissues. In support of the gene expression data, iWAT and BAT PDSS2[BKO] tissue pieces both exhibited increased ECAR rates compared to PDSS2[FL] (Fig. 7C; Appendix Fig. S2C), suggesting an increase in ATP production to support the upregulation of heat producing, futile cycling pathways. Other peripheral tissues such as liver, muscle, kidney, eWAT and heart also had increased gene expression of UCP1-independent thermogenic pathways. Notably, AAC1 or AAC2 gene expression was increased in all the tested peripheral tissues (Appendix Fig. S2G–K).

Lastly, we wanted to understand FGF21's mitohormetic effects on glucose metabolism in peripheral tissues of PDSS2[BKO] mice. Interestingly, Glut1, a known FGF21 mediator (Ge et al, 2011) had increased expression in BAT, eWAT, liver and muscle (Fig. 7I). Further, FGF21 seemed to have a mitohormetic effect on liver of PDSS2[BKO] mice fed a high fat diet. In line with studies that have shown that FGF21 increases hepatic gluconeogenesis (Liang et al, 2014), gene expression of G6PC and PCK1 was increased. Additionally, the ratio of spliced to unspliced XBP1, a readout for ER stress that is usually activated in obese livers, was decreased (Fig. 7J).

These findings reveal a mechanism within brown adipocytes where CoQ deficiency leads to the export of mtRNAs from the mitochondria, initiating the ISR through eIF2α phosphorylation by PKR. This phosphorylation of eIF2α subsequently causes increased transcription of stress genes, notably ATF4, resulting in elevated FGF21 expression. These data indicate circulating FGF21, secreted from CoQ-deficient BAT, exerts systemic metabolic effects, allowing PDSS2[BKO] animals to maintain a lean phenotype. Central to this observed metabolic adaptation is the ATF4-mediated ISR signaling. Furthermore, increases in oxygen consumption rates due to beiging in iWAT might explain the observed increased respiration rates in the PDSS2[BKO] animals in the face of UCP1 suppression. Additionally, upregulated pathways in other peripheral tissues could offer a mitohormetic advantage, as illustrated in Fig. 7K.

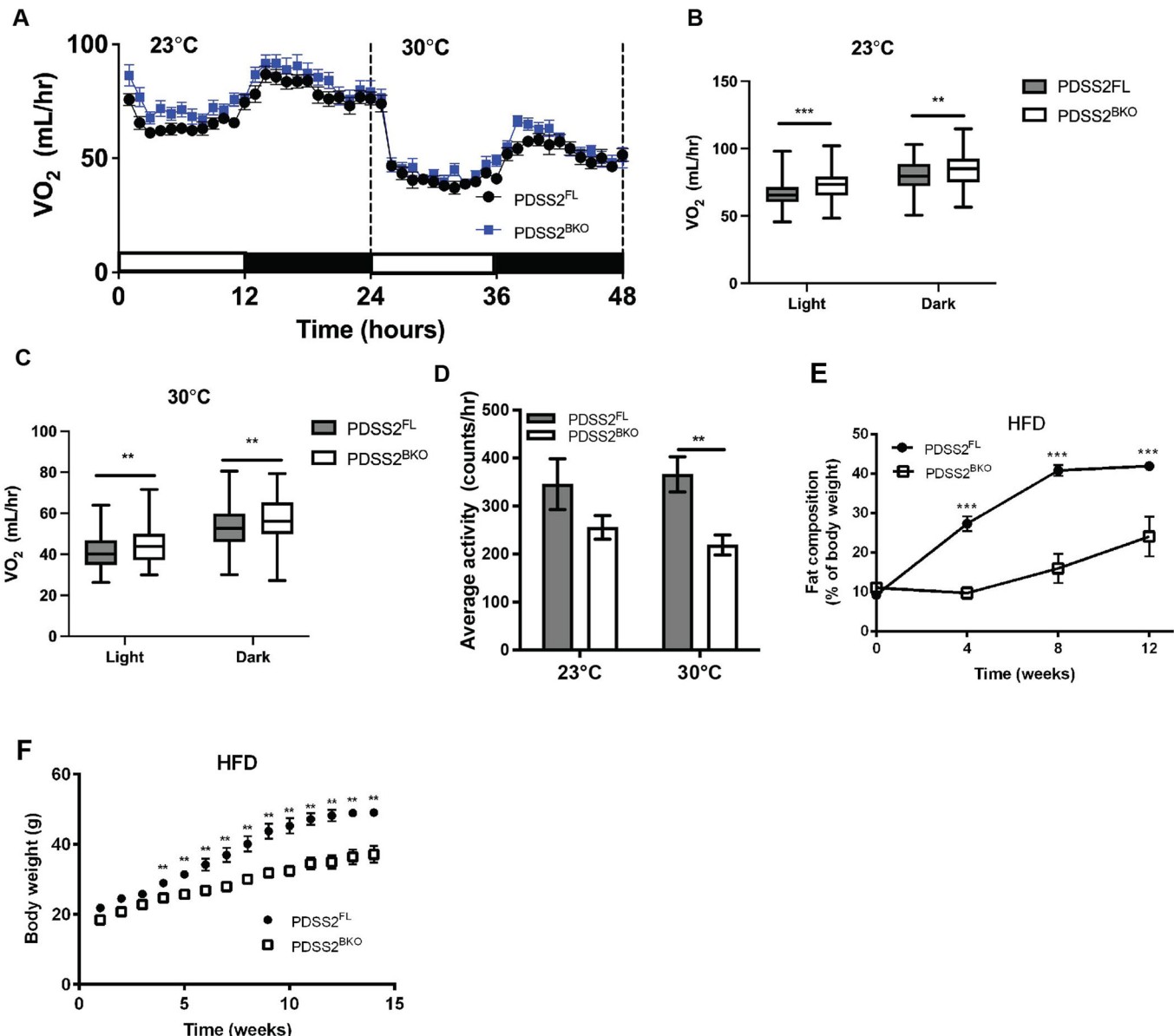

**Figure 5. BAT CoQ deficiency modulates whole body metabolism.**

(A) Oxygen consumption rate of PDSS2$^{FL}$ or PDSS2$^{BKO}$ animals was monitored under 23 and 30 °C in real-time in metabolic cages using CLAMS. (B, C) Average oxygen consumption rate. Centerline represents median and box extends from 25th to 75th percentiles. (D) Average ambulatory activity in 24 h at 23 and 30 °C. (E, F) Body weight and fat compositions were monitored during 12 weeks of HFD feeding. Data information: (A–D) $n = 10$–11/group. (E, F) $n = 4$/group. (A, D–F) Data are mean ± SEM. (B–F) Results were compared using an unpaired two-tailed Student's $t$ test. Significance presented at **$P < 0.01$, and ***$P < 0.001$ compared to controls. Data describes biological replicates. Source data are available online for this figure.

## Discussion

In previous studies, we established that coenzyme Q (CoQ) deficient brown adipocytes displayed lowered basal and uncoupled respiration due to suppressed uncoupling protein 1 (UCP1) expression (Chang et al, 2022). However, the mechanism by which CoQ regulates UCP1 expression remained unclear. Therefore, to further understand the downstream effects of CoQ depletion that lead to decreased UCP1 expression, we conducted RNA sequencing (RNAseq) analysis on murine brown adipocyte cells treated with

4CBA. This provided a comprehensive view of the transcriptional changes caused by CoQ deficiency. Intriguingly, the RNAseq analysis revealed an upregulation of both the integrated stress response (ISR) and the mitochondrial unfolded protein response (UPR$^{mt}$) genes (Figs. 1B–D and EV1D). This in-vitro upregulation was consistent with findings in BAT of mice with a UCP1-cre driven knockout of the CoQ biosynthetic protein PDSS2 (PDSS2$^{BKO}$) (Fig. 1F,G). Key observations from the 4CBA pharmacological treatment and PDSS2$^{BKO}$ mice were further corroborated by a genetically induced CoQ deficiency model using

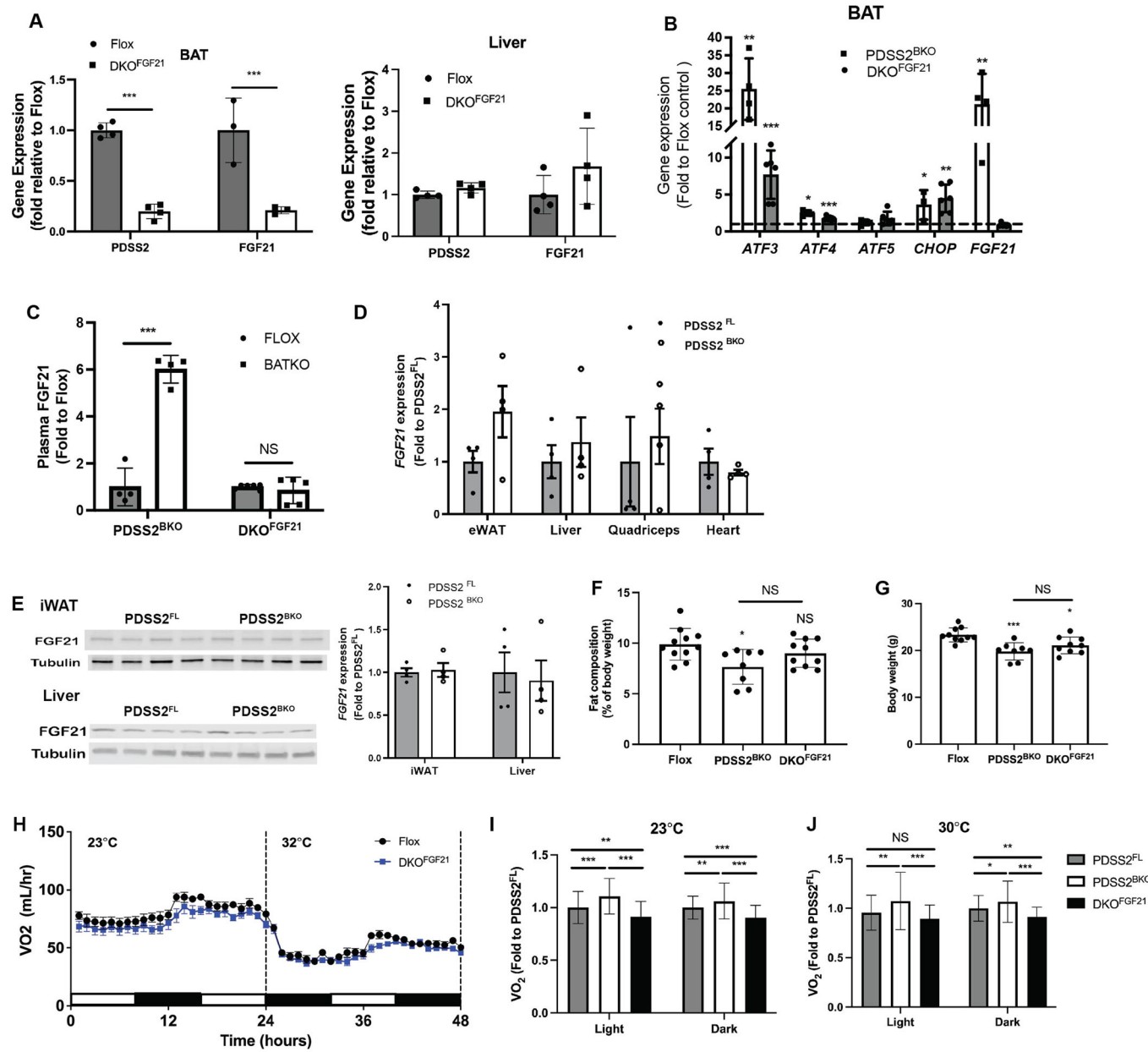

**Figure 6. FGF21 is crucial in metabolic adaptation by BAT CoQ deficiency.**

(A) Validation of tissue-specific knockout of PDSS2 and FGF21 via gene expression analysis in BAT (left panel) and Liver (right panel) of floxed controls (Flox) or BAT-specific PDSS2 and FGF21 double knockout (DKO$^{FGF21}$) animals, $n = 3$–4. (B) ISR-related gene expression in BAT of PDSS2$^{BKO}$ or DKO$^{FGF21}$, $n = 4$–6. (C) Plasma FGF21 levels of PDSS2$^{BKO}$ and DKO$^{FGF21}$ compared to floxed controls, $n = 4$–6. (D) FGF21 gene expression in tissues of PDSS2$^{BKO}$, $n = 4$. (E) FGF21 protein levels in Liver or iWAT of PDSS2$^{FL}$ and DKO$^{FGF21}$, $n = 4$. (F, G) Body weight and fat content of Flox, PDSS2$^{BKO}$ and DKO$^{FGF21}$ after 2-week defined diet at age 10 weeks, $n = 8$–11. (H) Oxygen consumption rate of PDSS2$^{FL}$ and DKO$^{FGF21}$ animals was monitored under 23 and 30 °C in real-time in metabolic cages using CLAMS, $n = 6$. (I, J) Averaged VO$_2$ of PDSS2$^{FL}$, PDSS2$^{BKO}$ and DKO$^{FGF21}$ at 23 °C or 30 °C, $n = 6$–11. Data information: Data are mean ± SEM. (A–E) Results were compared using an unpaired two-tailed Student's $t$ test. (F, G, I, J) Results were compared using a one-way ANOVA test. Significance presented at *$P < 0.05$, **$P < 0.01$, and ***$P < 0.001$ compared to controls. Data describes biological replicates. Source data are available online for this figure.

siRNA targeting COQ2, a crucial enzyme in COQ biosynthesis and the target of 4CBA. Alongside UCP1 downregulation (Figure EV2B,C), we noted increased expression of stress-related genes (Figure EV2B,D), and a decrease in respiration rates, irrespective of the substrate provided (Figure EV2E–H). Interestingly, this stress response was not observed in a liver-specific CoQ deficiency animal

strain (PDSS2$^{AlbKO}$) (Appendix Fig. S1A), suggesting the UPR$^{mt}$/ISR induction by CoQ deficiency is tissue-specific (Peng et al, 2008).

Given the intricate relationship between mitochondrial dynamics and cellular stress conditions, we wanted to elucidate potential changes in mitochondria structure. CoQ deficient brown adipocytes displayed elongated and tubular mitochondria

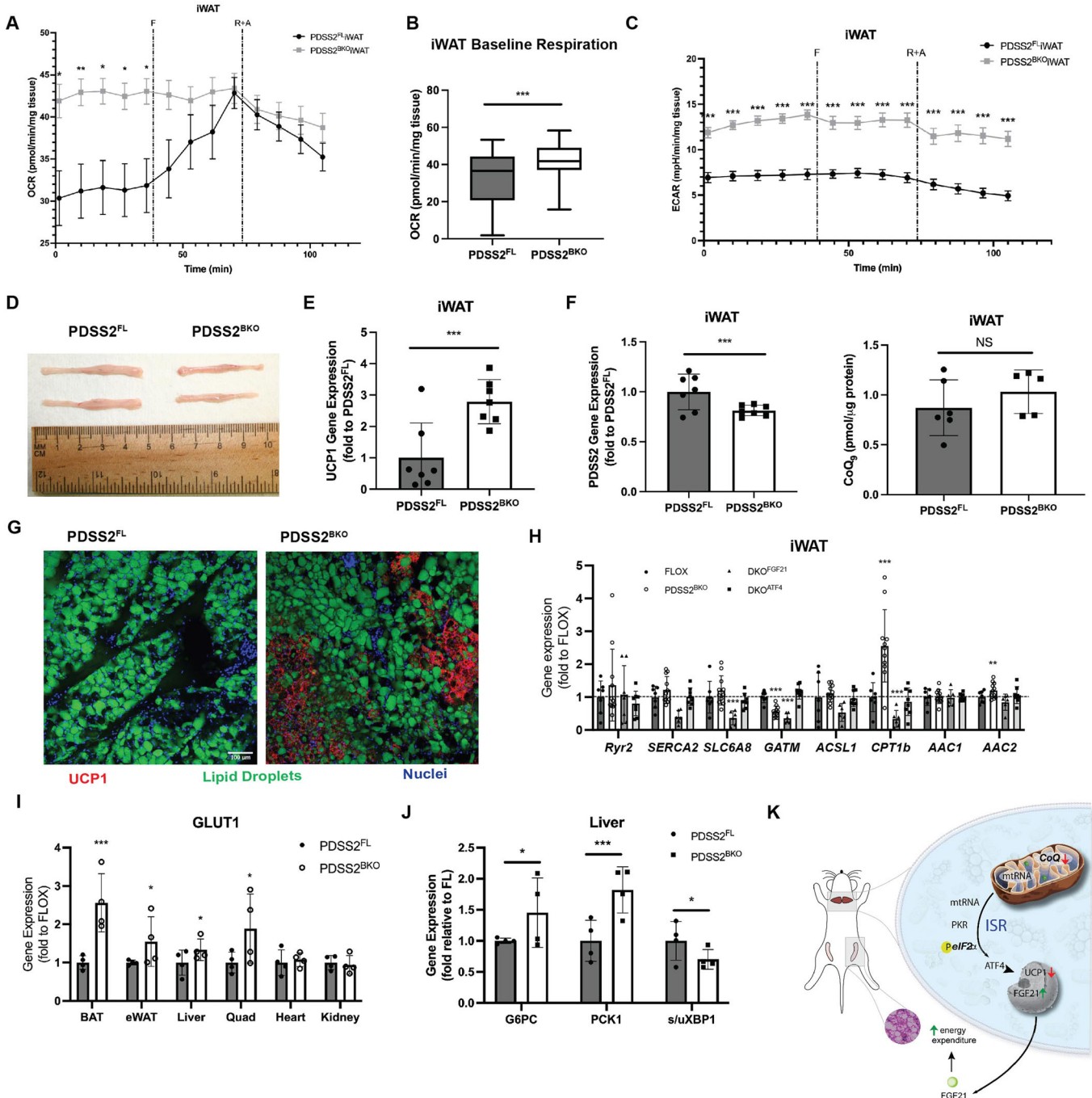

(Fig. 2A,B), impaired mitochondrial fission, and decreased phosphorylation of Drp1 (Fig. 2C). From ultrastructural analysis of mitochondria, we observed cristae hypoplasia and impaired cristae formation (Fig. 2D,E). Lipid droplets also appeared to be larger in CoQ-deficient brown adipocytes (Fig. 2A), although more in depth characterization is needed. Since it is known that CoQ is an essential cofactor for β-fatty oxidation (Liang et al, 2014), CoQ deficiency may result in impaired β-fatty acid oxidation and decreased utilization of lipids, leading to an accumulation of free fatty acids (FFAs) and a consequent increase in lipid droplet size. This is in line with the lower palmitate oxidation observed in OCR

data from the COQ2 siRNA model in comparison to control cells (Figure EV2G,H). Moreover, our previous study reported enlarged lipid droplets in BAT from PDSS2^BKO animals, displaying a whitening phenotype (Chang et al, 2022). The physiological impact of mitochondrial fragmentation on a cell or tissue can have varying effects based on the specific type of cell, particularly for highly metabolic tissues such as BAT (Sprenger and Langer, 2019; Wikstrom et al, 2014). It has been hypothesized that such network rearrangement could help protect against further damage by enhancing the resistance of mitochondria through the exchange of essential components. This is known as the stress-induced

◄ **Figure 7.  BAT CoQ deficiency induces UCP1-independent thermogenesis in peripheral tissues.**

(A) OCR of iWAT tissue pieces from PDSS2$^{FL}$ or PDSS2$^{BKO}$ mice. iWAT was dissected from 5 mice per group with 5 tissue pieces analyzed per mouse for a total of 25 tissue pieces analyzed per group. (B) Baseline respiration of iWAT tissue pieces. Data from the first 5 timepoints from the OCR trace in A were pooled and compared between PDSS2$^{FL}$ or PDSS2$^{BKO}$ iWAT pieces. Centerline represents median and box extends from 25$^{th}$ to 75$^{th}$ percentiles. iWAT was dissected from 5 mice per group with 5 tissue pieces analyzed per mouse for a total of 25 tissue pieces analyzed per group. (C) Extracellular acidification rate (ECAR) of iWAT tissue pieces from PDSS2$^{FL}$ or PDSS2$^{BKO}$ mice. iWAT was dissected from 5 mice per group with 5 tissue pieces analyzed per mouse for a total of 25 tissue pieces analyzed per group. (D) Representative image of iWAT lobes dissected from PDSS2$^{FL}$ or PDSS2$^{BKO}$ mice for size comparison. (E) UCP1 gene expression in iWAT of PDSS2$^{FL}$ or PDSS2$^{BKO}$ mice, $n = 7$. (F) Left panel: PDSS2 gene expression in iWAT PDSS2$^{FL}$ or PDSS2$^{BKO}$ mice, $n = 7$. Right Panel: CoQ$_9$ levels in iWAT PDSS2$^{FL}$ or PDSS2$^{BKO}$ mice, $n = 5$–6. (G) Immunohistochemistry analysis of iWAT tissue sections from PDSS2$^{FL}$ or PDSS2$^{BKO}$ mice. UCP1 protein is visualized in red, lipid droplets are visualized in green and nuclei are visualized in blue. Images are representative of sections from 3-4 animals per group. Scale bar = 100 μm. (H) Genes of UCP1-independent thermogenesis in iWAT of BAT-specific PDSS2, ATF4 and PDSS2, or FGF21 and PDSS2 knockouts (PDSS2$^{BKO}$, DKO$^{ATF4}$ and DKO$^{FGF21}$), $n = 6$–8/group. (I) Glut1 gene expression in various tissues of PDSS2$^{FL}$ or PDSS2$^{BKO}$ mice, $n = 4$. (J) G6PC, PCK1 and spliced/unspliced XBP1 (s/uXBP1) expression in liver from PDSS2$^{FL}$ or PDSS2$^{BKO}$ mice fed a high fat diet for 12 weeks $n = 4$. (K) Graphic scheme illustrating the mechanisms of mitohormetic responses by CoQ deficiency in BAT. Data information: (A–C, E, F, and H–J) Data are mean ± SEM. Results were compared using an unpaired two-tailed Student's $t$ test. Significance presented at *$P < 0.05$, **$P < 0.01$, and ***$P < 0.001$ compared to controls. Data describes biological replicates. Source data are available online for this figure.

mitochondrial hyperfusion (SIMH) pathway (Tondera et al, 2009). Moreover, in accordance with our finding, the observed fragmentation loss could be due to reduced UCP1 levels and uncoupling activity in 4CBA treated cells (Pisani et al, 2018). Considering these previous observations, we hypothesized that CoQ deficiency might induce sublethal effects, leading to an adaptive reconfiguration of the mitochondrial network. Supporting this mitochondrial quality control strategy, RNA sequencing data indicated an upregulation of several mitochondrial chaperones (Fig. 1A,B). Mitochondria can use internal protein chaperones and proteases to mitigate proteotoxic stress by refolding or degrading misfolded proteins (Fiorese and Haynes, 2017) and can also degrade mislocalized mitochondrial proteins that fail to be imported.

To deepen our understanding of the molecular signatures downstream of CoQ deficiency leading to ISR upregulation (Fig. 3A), we identified PKR as the stress kinase responsible for ISR induction during CoQ deficiency (Fig. 3B,C). We observed an enrichment of mitochondrial RNA (mtRNA) in the cytosol, suggesting an increased export of mtRNAs in CoQ deficient cells, a mechanism that has been shown to activate PKR (Sud et al, 2016; Fig. 3D). Given that mitochondrial stress response events have a sequential order on the transcriptional level (Bar-Ziv et al, 2020; Forsstrom et al, 2019), we strived to identify the subsequent activations of ISR and UPR$^{mt}$ key proteins. ISR has been evaluated as an essential element to UPR$^{mt}$ activation in response to mitochondria stressors in mammals. Our observations uncovered a well-coordinated mechanism, with eIF2α phosphorylation and ATF4 protein accumulation occurring after 7 h of 4CBA treatment (Fig. 4B,C). Gene expression of other factors involved in ISR and UPR$^{mt}$ were induced or peaked subsequently (Fig. 4D–F), followed by UCP1 expression suppression (Fig. 4G,H). In line with our findings, other publications have reported that UCP1 and PGC1α expression is increased in the BAT of ATF4-whole body knockout mice, suggesting that ATF4 can act as a negative regulator of uncoupled respiration (Wang et al, 2010).

Next, we turned our attention to how the induction of the stress response and the expression of factors such as FGF21 affect whole body metabolism during CoQ deficiency. We assessed FGF21 levels in both in vitro and in vivo models and measured the respiration of PDSS2$^{BKO}$ animals in metabolic cages. PDSS2$^{BKO}$ animals displayed increased oxygen consumption compared to floxed controls (Fig. 5A–C), offering protection against diet induced obesity (Fig. 5E,F). This increased respiration and protection from diet

induced obesity was dependent on BAT-derived FGF21 secretion (Fig. 6). Since FGF21 represents one of the most susceptible targets of ATF4, with three amino acid response elements (AARE) in its promoter region (Maruyama et al, 2016), we also found that upstream induction of ATF4 expression (Appendix Fig. S1) was needed to induce increased energy expenditure in PDSS2$^{BKO}$ mice. Although liver is thought to be the primary source of FGF21 production, it can be synthesized and secreted from other tissues during stress responses as well (Forsstrom et al, 2019). Indeed, diverse tissue-specific stimuli driving FGF21 expression are accompanied by a range of metabolic repercussions on distinct organs, involving varied signaling pathways and functional outcomes, as described by Spann et al (2021). Importantly, these data provide clear evidence that BAT-derived FGF21 can significantly contribute to circulating FGF21 levels and metabolic responses.

To determine the metabolic repercussion of FGF21, we next explored what pathways may be leading to the observed resistance to diet induced obesity in PDSS2$^{BKO}$ mice. Similar to the PDSS2$^{BKO}$ mice, another study found that genetically engineering liver, adipose tissue, or skeletal muscle to secrete FGF21 using adeno-associated virus (AAV), prevented weight gain and insulin resistance associated with aging (Jimenez et al, 2018). Recent reports also showed that ATF4 is required for metabolic adaptation in the BAT-specific OPA1 and BAT-specific Lrpprc knockout animals (Paulo et al, 2021; Pereira et al, 2021), illustrating the essential roles of ISR in metabolic adaptation. Interestingly, UCP1KO animals also exhibit protection from diet-induced obesity due to increased shivering to compensate for a lack of UCP1 (Liu et al, 2003). Although PDSS2$^{BKO}$ animals have decreased UCP1 expression, they exhibit increased respiration unlike UCP1KO animals. This suggested that perhaps the animals increased non-shivering thermogenic pathways to compensate for a lack of UCP1. To assess the autocrine role of FGF21, we first studied metabolic alterations to BAT. Although we do observe increased gene expression of UCP1-independent thermogenic pathways in BAT (Appendix Fig. S2D), BAT tissue pieces isolated from PDSS2$^{BKO}$ animals displayed decreased respiration rates compared to floxed controls (Appendix Fig. S2A,B), suggesting that BAT is not the primary organ responsible for the enhanced respiration in PDSS2$^{BKO}$ mice.

This prompted us to study peripheral organs to understand the downstream mitohormetic effects of BAT-CoQ deficiency. Inter-organ mitohormesis has been well studied in C. elegans, with one

study reporting a propagation of the UPR$^{mt}$ from neurons to the periphery in C. elegans, leading to increased longevity (Durieux et al, 2011). This mitohormetic interorgan communication has only recently been explored in mammals. Interestingly, one recent study showed that small extracellular vesicles (sEVs) released from adipocytes in response to an artificial mitochondrial stress can activate a protective pathway in cardiomyocytes, exemplifying inter-organ mitohormesis in mammalian systems (Crewe et al, 2021). Notably, several studies have highlighted white adipose tissue (WAT) and the liver as main targets of systemic FGF21 effects. This led us to hypothesize that these organs might be responsible for the increased respiratory rate observed in PDSS2$^{BKO}$. While primary hepatocytes isolated from PDSS2$^{BKO}$ liver did not have significant differences in respiration compared to controls (Appendix Fig. S2E,F), we detected metabolic remodeling in WAT, evidenced by higher oxygen consumption rates (OCR) in freshly isolated inguinal white adipose tissue (iWAT) from PDSS2$^{BKO}$ mice compared to iWAT from floxed controls (Fig. 7A,B). Numerous studies (Abu-Odeh et al, 2021; Fisher et al, 2012; Machado et al, 2022) have shed light on FGF21's role in white adipose tissue browning and metabolic regulation. We observed increased iWAT browning in PDSS2$^{BKO}$ mice, which we had briefly reported in the past (Chang et al, 2022). Here, we report increased gene expression of UCP1, and immunohisto-chemistry analysis revealed increased protein expression of UCP1, localized around adipocytes with a multilocular lipid droplet morphology (Fig. 7E,G). Other mitohormetic effects of FGF21 on peripheral tissues (Chen et al, 2017),were also observed, such as increased gene expression of UCP1-independent thermogenesis in iWAT, liver, muscle, heart, kidney and eWAT (Fig. 7H; Appendix Fig. S2G–K), and changes to glucose metabolism (Fig. 7I,J). Overall, the data present a unique model whereby a signal derived from dysfunctional BAT results in interorgan crosstalk, particularly between BAT and iWAT, resulting in mitohormetic effects on whole-body metabolism.

Although we identify FGF21 as a stress-induced factor required for potential crosstalk between BAT and WAT, unraveling the global contribution and activity of FGF21 as a distal signal in every organ remains challenging and still poses several knowledge gaps. Some studies have reported FGF21's direct action on adipose tissue (Samms et al, 2016), whereas other studies propose that FGF21 acts through the central nervous system (CNS) to assert metabolic benefit (Bar-Ziv et al, 2020; Liang et al, 2014). Thus, it remains uncertain whether FGF21 acts directly or indirectly through the CNS in the proposed model. Additionally, other metabolic modulators secreted from CoQ-deficient BAT might also influence systemic metabolic alteration, such as GDF-15 (Xiao et al, 2022) or extracellular vesicles (Crewe et al, 2021). Future research will focus on identifying contributions of other organs and further secreted factors involved in interorgan communication within the proposed model and the mechanisms by which these signals are propagated. Overall, the unexpected consequences of BAT CoQ deficiency, including a seemingly paradoxical anti-obesity response, are likely to have important implications for our understanding of CoQ-related disorders and previously unknown BAT-dependent effects of statin therapy. Furthermore, comparing various stressors and their responses could be crucial for understanding tissue susceptibility to stress and interorgan adaptation.

## Methods

For reagents and tools, please see Table 1.

**Table 1. Reagents and tools.**

| Reagent/resource | Source | Identifier/catalog number/sequence |
| --- | --- | --- |
| rt-qPCR primers | | |
| Ucp1 | Integrated DNA Technologies | Mm.PT.58.7088262 |
| Mthfd2 | Integrated DNA Technologies | Mm.PT.58.43172230 |
| Gdf15 | Integrated DNA Technologies | Mm.PT.58.13112185 |
| Psat1 | Integrated DNA Technologies | Mm.PT.58.21738493 |
| Phgdh | Integrated DNA Technologies | Mm.PT.58.10927981 |
| Atf3 | Integrated DNA Technologies | Mm.PT.58.41654515 |
| Atf4 | Integrated DNA Technologies | Mm.PT.58.31755577 |
| Atf5 | Integrated DNA Technologies | Mm.PT.58.13930057 |
| Chop | Integrated DNA Technologies | Mm.PT.58.30882054 |
| Fgf21 | Integrated DNA Technologies | Mm.PT.58.29365871.g |
| Lonp1 | Integrated DNA Technologies | Mm.PT.58.8021338 |
| Ppia | Integrated DNA Technologies | Mm.PT.39a.2.gs |
| Hsp60 | Integrated DNA Technologies | Mm.PT.58.13557954 |
| Clpp | Integrated DNA Technologies | Mm.PT.58.7137305 |
| Bip | Integrated DNA Technologies | Mm.PT.58.6115287.g |
| COQ2 | Integrated DNA Technologies | Mm.PT.58.9367270 |
| PDSS2 | Integrated DNA Technologies | Mm.PT.58.9704081 |

**Table 1.**  (continued)

| Reagent/resource | Source | Identifier/catalog number/sequence |
|---|---|---|
| UCP1 | Integrated DNA Technologies | Hs.PT.58.39157006 |
| PPIA | Thermo Fisher Scientific | Hs01565700_g1 |
| HSP60 | Thermo Fisher Scientific | Hs01036753_g1 |
| ATF4 | Thermo Fisher Scientific | Hs00909569_g1 |
| ATF5 | Thermo Fisher Scientific | Hs00247172_m1 |
| FGF21 | Thermo Fisher Scientific | Hs00173927_m1 |
| CHOP | Thermo Fisher Scientific | Hs99999172_m1 |
| LONP1 | Thermo Fisher Scientific | Hs00998407_m1 |
| ND1 | Integrated DNA Technologies | N/A (Custom)<br>Forward: CTAGCAGAAACAAACCGGGC<br>Reverse: CCGGCTGCGTATTCTACGTT |
| ND4 | Integrated DNA Technologies | N/A (Custom)<br>Forward: CAACCAAACTGAACGCCTAAAC<br>Reverse: GAGGGCAATTAGCAGTGGAATA |
| ND5 | Integrated DNA Technologies | N/A (Custom)<br>Forward: CGGAGCCCTAACCACATTATT<br>Reverse: GCCTAGTTGGCTTGATGTAGAG |
| CYTB | Thermo Fisher Scientific | Mm04225271_g1 |
| Ryr2 | Thermo Fisher Scientific | Mm00465877_m1 |
| SDHB | Thermo Fisher Scientific | Mm00458272_m1 |
| Serca2 | Integrated DNA Technologies | Mm.PT.58.5303089 |
| Slc6a8 | Integrated DNA Technologies | Mm.PT.58.6422914 |
| Gatm | Integrated DNA Technologies | Mm.PT.56a.42547208 |
| Acsl1 | Integrated DNA Technologies | Mm.PT.58.29312796 |
| Cpt1b | Integrated DNA Technologies | Mm.PT.58.2340927 |
| Aac1 | Integrated DNA Technologies | Mm.PT.58.30852882 |
| Aac2 | Integrated DNA Technologies | Mm.PT.58.32860004 |
| Glut1 | Integrated DNA Technologies | Mm.PT.58.7590689 |
| G6pc1 | Integrated DNA Technologies | Mm.PT.58.11964858 |
| Pck1 | Integrated DNA Technologies | Mm.PT.5811992693 |
| sXBP1 | Integrated DNA Technologies | N/A (Custom)<br>Forward: GCTGAGTCCGCAGCAGGT<br>Reverse: CAGGGTCCAACTTGTCCAGAAT |
| uXBP1 | Integrated DNA Technologies | N/A (Custom)<br>Forward: CAGACTACGTGCACCTCTGC<br>Reverse: CAGGGTCCAACTTGTCCAGAAT |
| Antibodies | | |
| ATF4 | Cell Signaling Technology | Cat# 11815, RRID:AB_2616025 |
| Phospho-eIF2α (Ser51) | Cell Signaling Technology | Cat# 3398, RRID:AB_2096481 |
| eIF2α | Cell Signaling Technology | Cat# 9722, RRID:AB_2230924 |
| Phospho-Drp1 (Ser616) | Cell Signaling Technology | Cat# 3455, RRID:AB_2085352 |
| Drp1 | Abcam | Cat# ab184247 |
| VDAC1/Porin | Abcam | Cat# ab154856, RRID:AB_2687466 |
| FGF21 | Abcam | Cat# ab171941, RRID:AB_2629460 |
| beta Tubulin | Abcam | Cat# ab131205, RRID:AB_11156121 |
| beta Actin | Sigma-Aldrich | Cat# A1978, RRID:AB_476692 |
| ATF3 | Abcam | Cat# ab207434, RRID:AB_2734728 |
| UCP1 | Abcam | Cat# ab209483 RRID:AB_2722676 |

**Table 1.** (continued)

| Reagent/resource | Source | Identifier/catalog number/sequence |
| --- | --- | --- |
| UCP1 (E9Z2V) XP Rabbit mAb | Cell Signaling Technology | Cat# 72298 |
| PKR (B-10) | Santa Cruz | Cat#sc-6282 RRID:AB_628150 |
| Phospho-PKR (Thr451) | Thermo Fisher Scientific | Cat#44-668G RRID:AB_2533716 |
| GCN2 | Cell Signaling Technology | Cat# 3302, RRID: AB_2277617 |
| PERK (C33E10) Rabbit mAb | Cell Signaling Technology | Cat# 3192, RRID: AB_209584 |
| IRDye 680LT Goat anti-Mouse IgG antibody | LI-COR Biosciences | Cat# 925-68020, RRID:AB_2687826 |
| IRDye 800CW Goat anti-Rabbit IgG antibody | LI-COR Biosciences | Cat# 925-32211, RRID:AB_2651127 |
| Alexa Fluor 594 nm Donkey anti-Rabbit IgG antibody | Jackson ImmunoResearch Laboratory Inc. | Cat# 711-585-152, RRID:AB_2340621 |

## Mouse model

All animal experiments were approved and performed under the guidelines and ethical regulations established by the UC Berkeley Animal Care and Use Committee under the protocol ID AUP-2019-05-12180-1. The animals were housed in an AAALAC-certified facility and supervised by licensed technicians. They were monitored daily by the Office of Laboratory Animal Care (OLAC) veterinary lab staff for any health conditions during the course of the experiments. Brown-fat specific PDSS2 knockout (PDSS2[BKO]) mice were generated by crossing PDSS2-LoxP mice (gift from Dr. Gasser) (Galmozzi et al, 2014) with UCP1-cre mice (mouse model developed by the Rosen lab deposited in JAX, stock #024670) on a C57BL/6J background. Liver-specific PDSS2 knockouts (PDSS2[AlbKO]) were also generated by crossing PDSS2-LoxP mice with Albumin-cre mice (Postic et al, 1999) (JAX, stock # 003574). Animals were fed a regular chow diet (LabDiet 5053) under the ambient temperature at 23 °C. Male animals aged between 6 and 8 weeks old were fed defined diet (Research diet D12450J) for two to three weeks before experiments, unless described otherwise. The defined diet contains only 32% of CoQ content compared to standard chow; thus, it is suitable in our experiments for limiting CoQ intake. Diet-induced obesity was performed on male animals at 6 weeks old. Mice were fed with a 60 kcal% fat diet (Research diet D12492) for 12 weeks. Food consumption was monitored once per week, and accumulated food consumption was shown as accumulated amount over three weeks per animal. PDSS2 flox allele control animals (PDSS2[FL]) were used as control relative to PDSS2[BKO]. Brown fat-specific PDSS2 and FGF21 double knockout (DKO[FGF21]) mice were generated by crossing PDSS2[BKO] with FGF21-LoxP mice (Potthoff et al, 2009) (JAX stock #022361). Brown fat-specific PDSS2 and ATF4 double knockout DKO[ATF4] were generated by crossing PDSS2[BKO] with ATF4-LoxP mice (Ebert et al, 2012). Flox/Flox littermates were used as control, respectively. All animal studies were performed using age-matched male mice (8–12 weeks), cohorts greater than or equal to five mice per genotype or treatment and repeated at least twice.

## Cell cultures

### Immortalized murine brown adipocytes

The immortalized murine brown preadipocyte line is a generous gift from Shingo Kajimura (Galmozzi et al, 2014). Cells were cultured in DMEM supplemented with 10% FBS, 1% penicillin, and streptomycin. Murine brown adipocytes were prepared from adipogenic differentiation of the confluent preadipocytes with DMEM containing 10% FBS, penicillin/streptomycin (P/S) and differentiation cocktails (850 nM insulin, 2 nM T3, 125 nM indomethacin, 0.5 mM 3-isobutyl-1-methylxanthine, 1 µM dexamethasone, and 1 µM rosiglitazone) for 2 days. Two days after induction, cells were switched to a maintenance medium containing 10%FBS, P/S, insulin, T3 and rosiglitazone. Experiments assessing CoQ deficiency on thermogenic function were performed on day 4–6 of differentiation (48 h) or day 5–6 (24 h) in the presence of 4-chlorobenzoic acid (4CBA, 4 mM) or vehicle (0.1% DMSO and 1% ethanol in culture medium) for murine adipocytes. ISR inhibitors were added with 4CBA (4 mM) in the maintenance medium at day 5 of differentiation, for 24 h to assess UCP1 rescue effects. ISRIB (200 nM), GSK (1 µM), A92 (10 µM), Imoxin (2 µM), or 7DG (10 µM). The cell line is tested negative for mycoplasma contamination.

### Human beige cells

Human mesenchymal stem cells isolated from subcutaneous fats were purchased from ZenBio and cultured in DMEM/F12 supplemented with 10% FBS, 5 ng/ml bFGF and FGF, 10 mM HEPES, 1% penicillin and streptomycin. Beige adipogenesis was induced 2 days post confluence with DMEM/F12 (10% FBS, 10 mM HEPES, Pen/Strep) in the presence of the aforementioned differentiation cocktail for 3 days (Day0–Day3). Cells were then switched to DMEM/F12 (10% FBS, 10 mM HEPES, Pen/Strep) containing insulin and T3 for 5 days. 4 mM 4CBA or vehicle (0.1% DMSO and 1% ethanol in culture medium) were added at day 8 to day 10 of differentiation. The cell line is tested negative for mycoplasma contamination.

### Primary hepatocyte isolation

Primary hepatocytes were isolated from mouse livers as described in Parlakgül et al (2022). Briefly, mice were anesthetized using 300 mg/kg ketamine and 30 mg/kg xylazine in PBS. Perfusion of the liver was carried out through the portal vein using 50 mL of buffer I (11 mM Glucose, 200 µM EGTA, 1.17 mM $MgSO_4$ heptahydrated, 1.19 mM $KH_2PO_4$, 118 mM NaCl, 4.7 mM KCl, 25 mM $NaHCO_3$, pH 7.32) via an osmotic pump at a speed of 4 mL/min. After the liver turned pale, the pump speed was increased to 7 mL/min. 50 mL of buffer II (11 mM Glucose; 2.55 mM $CaCl_2$; 1.17 mM MgSO4 heptahydrated; 1.19 mM $KH_2PO_4$; 118 mM NaCl; 4.7 mM KCl; 25 mM $NaHCO_3$; BSA (bovin serum albumin) fatty acid free

7.2 mg/mL; 0.18 mg/mL of Liberase™ (Sigma)) was then pumped through the liver. Hepatocytes were carefully released and centrifuged at 500 rpm for 5 min. Hepatocytes were washed two times with 25 mL of Williams E Media containing 5% cosmic calf serum (CCS) and 1 mM glutamine. Hepatocytes were then layered on a 30% Percoll gradient and centrifuged at 1500 rpm for 15 min. The healthy cells were recovered at the bottom of the tube, resuspended in Williams E Media containing 5% CCS and 1 mM glutamine and plated at a density of ~30,000 cells/well in a 24-well SeaHorse plate coated with type I collagen (Sigma).

## siRNA-mediated knockdown of target genes

Scramble or predesigned siRNA targeted to mouse ATF4, ATF5, CHOP, COQ2 or HRI (MISSION siRNA, MilliporeSigma) was diluted in Lipofectamine RNAiMAX reagent and applied on day 3 of differentiating adipocytes overnight (20 pmol of siRNA per one well of 12-well plate), and then cells were switched to a maintenance medium containing 10% FBS, P/S, insulin, T3 and rosiglitazone. siRNA universal negative controls were used as a control for silencing experiments (MISSION siRNA, Millipore-Sigma). For all siRNAs besides COQ2, 4CBA treatment occurred on day 5 to day 6 of differentiation.

## Mitochondria isolation

Mitochondria were isolated as previously described (Stefely and Pagliarini, 2017). Briefly, murine brown adipocytes were trypsinized, washed in PBS and resuspended in STE buffer (250 mM sucrose, 5 mM Tris and 2 mM EGTA, pH 7.4) with 1% BSA. Cells were homogenized using 10 strokes with a tight fit dounce homogenizer. Cells were then centrifuged at $8500 \times g$ for 10 min at 4 °C and the pellet was resuspended in STE with 1% BSA. Centrifugation at $700 \times g$ for 10 min at 4 °C was done to get rid of nuclei. The nuclei pellet was discarded, and the supernatant was transferred to a new tube and centrifuged at $8500 \times g$ for 15 min at 4 °C to pellet the crude mitochondrial fraction.

## Oxygen consumption measurements

### Tissue pieces

For assessing respiration in iWAT and BAT, the protocol by Bugge et al (2014) was followed with minor modifications. Briefly, freshly isolated mouse BAT and iWAT pieces were rinsed with XF-DMEM (Agilent) supplemented with 25 mM HEPES. The tissues were thoroughly cleaned of non-adipose material and blood vessels and cut into ~10 mg pieces. After repeated rinses, every piece was placed in the center of a well of XF24 Islet Capture microplate (#101122-100 Seahorse, Agilent) and then covered with the customized screens provided by the manufacturer. Immediately after, XF-DMEM, supplemented with 1 mM sodium pyruvate, 25 mM glucose and 1 mM glutamine, was added to each well. Then OCRs were analyzed with a Seahorse XFe24 Extracellular Flux Analyzer (Agilent, Santa Clara, CA, USA). To stimulate mitochondrial respiration in iWAT, carbonyl cyanide 4-(trifluoromethoxy) phenylhydrazone (FCCP; Sigma-Aldrich, St. Louis, MO), antimycin A (Sigma-Aldrich, St. Louis, MO) and rotenone (Sigma-Aldrich, St. Louis, MO) were injected at specific time points at a final concentration of 20 µM for each compound.

### Immortalized murine brown adipocytes

Oxygen consumption rate (OCR) was measured in differentiated brown adipocytes using the Seahorse XFe24. Brown adipocytes were cultured in a 24-well Seahorse plate. Before the experiments, cells were incubated in the XF assay medium supplemented with 10 mM glucose, 2 mM sodium pyruvate, and 2 mM GlutaMAX in a 37 °C $CO_2$ free incubator for an hour. Cells were subjected to the mitochondrial stress test by sequentially adding oligomycin (1 µM), FCCP (1 µM), and antimycin/rotenone mix (1 µM/1 µM).

To determine fatty acid oxidation, cells were treated overnight with substrate limitation medium (DMEM supplemented with glucose 0.5 mM, glutamine 1 mM, FBS 1% and L-carnitine 0.5 mM). Then, before the assay, cells were treated with XF Base medium supplemented with 2 mM glucose, 0.5 mM carnitine and Palmitate (200 µM) dissolved in BSA (6:1, Palmitate: BSA ratio) and incubated for 1 h at 37 °C without $CO_2$. Fatty acid oxidation test was performed with Seahorse XFe24 by sequentially adding Etomoxir (Sigma-Aldrich, St. Louis, MO) (4 µM) or medium, oligomycin (1 µM), FCCP (1 µM), and antimycin/rotenone mix (1 µM/1 µM). Measurements were normalized on protein content performed by BCA assay.

### Murine primary hepatocytes

Primary hepatocytes were isolated and seeded into type I collagen (Sigma) coated 24-well SeaHorse plates and allowed to attach overnight. Oxygen consumption rate (OCR) was measured using the SeaHorse XFe24. Before the experiments, cells were incubated in the XF assay medium supplemented with 20 mM glucose and 1 mM sodium pyruvate, in a 37 °C $CO_2$ free incubator for an hour. Cells were subjected to the mitochondrial stress test by sequentially adding oligomycin (2 µM), FCCP (1 µM), and antimycin/rotenone mix (2 µM/2 µM). Measurements were normalized on protein content performed by BCA assay.

## RNA preparation and quantitative RT-PCR

mRNA was extracted from tissues or in vitro cultures with TRIzol reagent (Invitrogen) and purified using Direct-zol RNA miniprep (R2055, Zymo) following the manufacturer's instruction with DNase treatment. Cytosolic RNA was extracted from in vitro murine brown adipocyte cells using the Cytoplasmic and Nuclear RNA purification Kit (21000, Norgen Biotek Corp.) following the manufacturer's instructions with DNase treatment. Mitochondrial RNA was extracted from isolated mitochondria fraction. Briefly, the mitochondrial pellet was resuspended in STE buffer and treated with 7 µL of RNase I (10 U/µL) (EN0601, Thermo Scientific) for 5 min on ice to remove remaining non-mitochondrial RNA. The mitochondrial suspension was then centrifuged at $8500 \times g$ for 20 min at 4 °C and the remaining pellet was resuspended in TRIzol reagent (15596018, Invitrogen) for downstream extraction of mitochondrial RNA using Direct-zol RNA miniprep (R2055, Zymo) with DNase treatment. cDNA was synthesized using Maxima First Strand cDNA synthesis kit (K1672, thermo scientific), and 10–20 ng cDNA was used for qPCR on a QuantStudio 5 real-time PCR system with either TaqMan Universal Master Mix II and validated PrimeTime primer probe sets (Integrated DNA Technologies), or Maxima SYBR Green/ROX qPCR Master Mix (2×) (K0221, Thermo Scientific) and designed primers. The ΔCt method was used to comparatively assess gene expression. Primer sequences are listed in the Reagents and Tools table.

## RNA-sequencing

Murine brown adipocytes were treated with/without 4CBA (4 mM) for 24 h, and $n = 3$ independent biological replicates were pooled to generate total RNA samples. Sequencing libraries were constructed from mRNA using KAPA mRNA HyperPrep Kit (KK8580) and NEXFlex barcoded adapters (Bioo Scientific). High-throughput sequencing was performed using a HiSeq 2500 instrument (Illumina) at the UC Berkeley QB3 Core. Raw reads were mapped using STAR (Dobin et al, 2013) against mouse (mm10) genome. DESeq2 (Love et al, 2014) was used to determine differential gene expression. Enriched transcription factor binding sites in gene promoters were identified using HOMER (Heinz et al, 2010). The RNAseq dataset produced in this study is available at the Gene Expression Omnibus (GEO) under accession GSE165940.

## Plasma FGF21

Plasma collected using EDTA was used for FGF21 determination by Mouse/Rat FGF-21 Quantikine ELISA Kit (R&D System) per the manufacturer's instruction.

## In vivo respirometry

Between 6- and 8-week-old male animals were fed a low-fat defined diet (Research Diets D12450J) under ambient temperature (23 °C) for 2 weeks. Food intake and body weight were monitored three times a week. After 2 weeks of low-fat defined diet feeding, body composition of animals were measured using an EchoMRI™-100V Body Composition Analyzer (EchoMRI™). Whole-body energy expenditure ($VO_2$, $VCO_2$) was recorded at indicated environmental temperatures using a Comprehensive Lab Animal Monitoring System (CLAMS, Columbus Instruments). Data were imported and analyzed using the web-based analysis tool for indirect calorimetry experiments (CalR) (Mina et al, 2018).

## Mitochondrial morphology assessment

Murine brown adipocytes were differentiated from immortalized preadipocytes and treated with 4CBA (4 mM) as previously described. Cells were incubated with Mito Tracker Deep Red (Thermo Fisher Scientific, Roskilde, Denmark) (100 nM) in culture media at 37 °C for 40 min. Then cells were washed twice with PBS and incubated with Hoechst (Thermo Fisher Scientific, Roskilde, Denmark) (2 µg/ml) and BODIPY 3922 (Thermo Fisher Scientific, Roskilde, Denmark) (1 µM), for nuclei and lipid droplets staining, respectively, for 10 min before imaging. Then, samples were washed twice in PBS and kept in FluoroBright DMEM (Gibco) supplemented with 1 mM glutamine. Microscope imaging was taken using ZEISS LSM 880 with a ×100/1.4NA oil immersion objective. The aspect ratio (ratio between centerline length and average width) and roundness were employed as mitochondrial morphometric parameters, according to Marchi et al (2017). FIJI standard tools were used to analyze images. At least five randomly chosen fields for each condition were analyzed. Representative images shown in the paper were cropped for detail using FIJI. To emphasize the fluorescent structures of mitochondria, representative images were separated into respective channels, the red channel was selected, and the fields were segmented using an otsu-thresholding method with identical parameters for each image.

For transmission electron microscopy, murine brown adipocytes were treated with 4 mM 4CBA for 48 h and fixed in FGP fixative (2.5% formaldehyde, 5% glutaraldehyde, 0.06% piric acid, 0.1 M cacodylate buffer, pH 7.4). Samples were rinsed in 0.1 M sodium cacodylate buffer, pH 7.2, and immersed in 1% osmium tetroxide with 1.6% potassium ferricyanide in 0.1 M sodium cacodylate buffer for 30 min. Samples were rinsed in buffer and briefly washed with distilled water (1×; 1 min, RT), then subjected to an ascending ethanol gradient followed by pure ethanol. Samples were progressively infiltrated with Epon resin (Electron Microscopy Sciences) and polymerized at 60 °C for 24–48 h. Following polymerization, the glass coverslips were removed using ultra-thin Personna razor blades (Electron Microscopy Sciences) and liquid nitrogen exposure, as needed. Regions of interest were cut and mounted on a blank resin block with cyanoacrylate glue for sectioning. Thin sections (80 nm) were cut using a Leica UC6 ultramicrotome (Leica, Wetzlar, Germany) from the surface of the block and collected onto formvar-coated 50 mesh grids. The grids were post-stained with 2% uranyl acetate followed by Reynold's lead citrate, for 5 min each. The sections were imaged using a FEI Tecnai 12 120 kV TEM (FEI) and data recorded using a Gatan Rio 16 CMOS with Gatan Microscopy Suite software (Gatan Inc). Fiji was used to analyze cristae formation. Three independent experiments were carried out for each treatment group with 50–60 mitochondria analyzed per independent experiment.

## Immunohistochemistry analysis of iWAT

iWAT was dissected, fixed in 4% paraformaldehyde at 4 °C overnight, cryopreserved in 30% sucrose at 4 °C overnight and embedded in Neg50. iWAT pieces were sectioned to generate 10 µm slices, washed with 1×PBS and blocked with a blocking buffer (HBSS fortified with: 10% FCS, 0.1% BSA, 0.05% saponin, and 2% donkey serum) for 1 h. Samples were incubated in UCP1 primary antibody (1:100) (Cell Signaling) diluted in blocking buffer overnight at 4 °C. Samples were washed with blocking buffer three times. Alexa Fluor 594 nm donkey anti-rabbit IgG secondary antibody (1:800) (Jackson ImmunoResearch Laboratory Inc.), 1.5 µg/mL Dapi and 3 µM BODIPY 3922 diluted in blocking buffer was applied to samples for 2 h at room temperature. Samples were mounted using Prolong Gold Antifade reagent (Cell Signaling) and imaged with a Zeiss LSM 710 confocal microscope.

## Glucose and insulin tolerance tests

Prior to a glucose tolerance test (GTT), mice were fasted with free access to water for 6 h. Prior to an insulin tolerance test (ITT), mice were fasted with free access to water for 4 h. Following the fast, a baseline glucose measurement is established by nicking the tail and collecting a small amount of blood onto a glucometer with glucose test strips. Then mice are injected intraperitoneally with either 2.0 g/kg body weight of glucose for a GTT or 0.4 IU/kg body weight of insulin for an ITT. Blood glucose measurements were then conducted 15, 30, 60, 90 and 120 min after the injection.

## Immunoblot

BAT mitochondria were isolated as previous described (Anderson et al, 2015), and protein lysates were extracted using RIPA lysis buffer. 25–30 µg of total cellular proteins were separated by gel electrophoresis on either a 4–20% gradient TGX gels or 7.5% SuperSep Phos-Tag Gel (Fujifilm Wako Chemicals) and transferred onto PVDF or nitrocellulose membranes using Trans-Blot Turbo Transfer system (BioRad). If using the SuperSep Phos-tag Gel, gels were incubated in transfer buffer with 10 mmol/L EDTA to remove $Zn^{2+}$ prior to transfer. Proteins were detected with the indicated antibodies listed in the Reagents and Tools table, and blots were imaged using Odyssey Imaging System. Image Studio Lite was used for protein quantification.

## Experimental study design

No sample-size calculation was performed to determine sample size, rather sample size was determined based on commonly used standards and experience in the field. For work with cell culture, a sample size of $n = 3–8$/group was used. For gene and protein expression in mouse tissue, a sample size of $n = 4–6$/group was used. For body weight, body composition and respiration measurements, in animals, a sample size of $n = 4–10$/group was used. Samples were randomized in in vitro studies. For in vivo experiments, only age-matched male mice per group were used in the study, and no other criteria were considered. For blinding, mice and samples were numbered independent of the genotypes.

## Statistical analysis

All measurements were taken from distinct samples. Data were analyzed using Prism and are expressed as mean ± SEM. Statistical significance was determined by unpaired two-tailed Student's *t* test or one-way ANOVA for the comparison of two conditions. Significance presented at $*P < 0.05$, $**P < 0.01$, and $***P < 0.001$ compared to controls or otherwise indicated.

# Data availability

RNA-sequencing data are available for public use at the Gene Expression Omnibus (GEO) under accession GSE165940. Other data that supports the findings of this manuscript can be found in this article or obtained from the corresponding author upon request.

# Peer review information

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

## Acknowledgements

This work was supported by NIH grant 1R01DK126830-01 to AS. ALG was supported by the American Heart Association predoctoral fellowship 23PRE1018992 (https://doi.org/10.58275/AHA.23PRE1018992.pc.gr.161143). GAT was supported by American Diabetes Association postdoctoral fellowship #1-19-PDF-058. We thank the UC Berkeley imaging core facility members Denise Schichnes, and Dr. Steve Ruzin for their support. We thank Dr. Danielle Jorgens and Reena Zalpuri at the UC Berkeley Electron Microscope Laboratory for advice and assistance in experiment planning, electron microscopy sample preparation, and data collection. We thank Dr. David L. Gasser for providing PDSS2-LoxP mice and Dr. Christopher M Adams for providing ATF4-LoxP mice.

## Author contributions

**Ching-Fang Chang**: Conceptualization; Formal analysis; Investigation; Visualization; Methodology; Writing—original draft. **Amanda L Gunawan**: Conceptualization; Formal analysis; Investigation; Visualization; Methodology; Writing—original draft; Writing—review and editing. **Irene Liparulo**: Formal analysis; Investigation; Visualization; Methodology; Writing—review and editing. **Peter-James H Zushin**: Investigation; Methodology. **Kaitlyn Vitangcol**: Investigation. **Greg A Timblin**: Formal analysis; Investigation. **Kaoru Sajio**: Resources; Supervision. **Biao Wang**: Resources; Supervision. **Güneş Parlakgül**: Software; Methodology. **Ana Paula Arruda**: Resources; Supervision; Investigation; Methodology. **Andreas Stahl**: Conceptualization; Supervision; Funding acquisition; Methodology; Writing—original draft; Project administration; Writing—review and editing.

**Disclosure and competing interests statement**
The authors declare no competing interests.

# Expanded View Figures

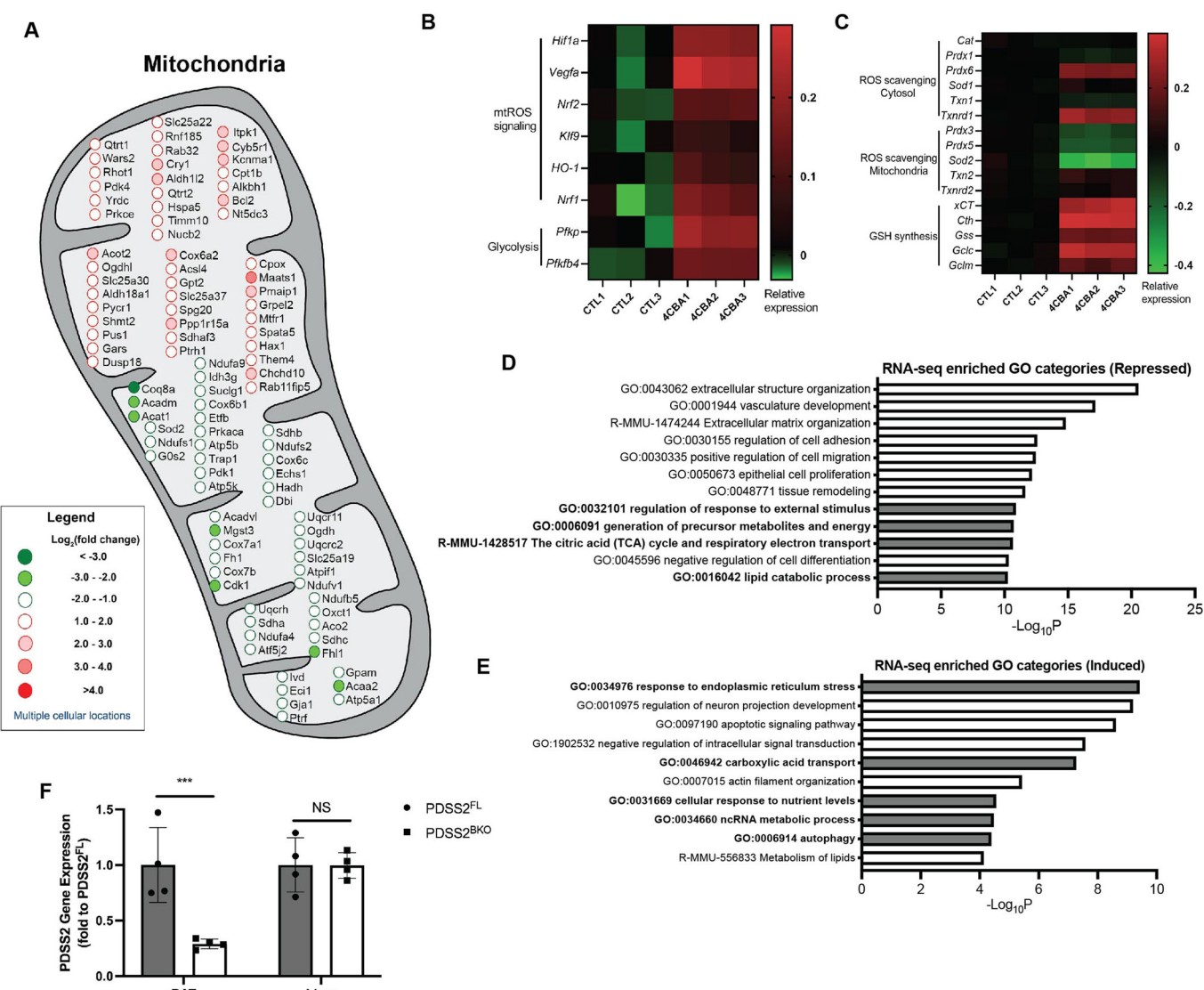

**Figure EV1. Enriched genes and pathways in CoQ deficient murine brown adipocytes.**

(A) Top 50 upregulated and top 50 downregulated mitochondrial genes from RNA sequencing results with *p* value ≤ 0.00418062. (B, C) Relative expression ($\log_{10}$-transformed RPKM value) of enriched genes in murine brown adipocytes treated with 4CBA (24 h). (D, E) Gene Ontology (GO) analysis of RNAseq data from 24-h 4CBA treated brown adipocytes. Representative GO categories repressed or induced. (F) Validation of tissue specific knockout of PDSS2 via qPCR analysis in BAT and liver of PDSS2 floxed (PDSS2^FL) or BAT-specific PDSS2 knockout (PDSS2^BKO) animals, *n* = 4. Results were compared using an unpaired two-tailed Student's *t* test. Significance presented at ***$P < 0.001$ compared to controls. Data describes biological replicates. Data information: (A, D, E) The Wald test was used for statistical analysis.

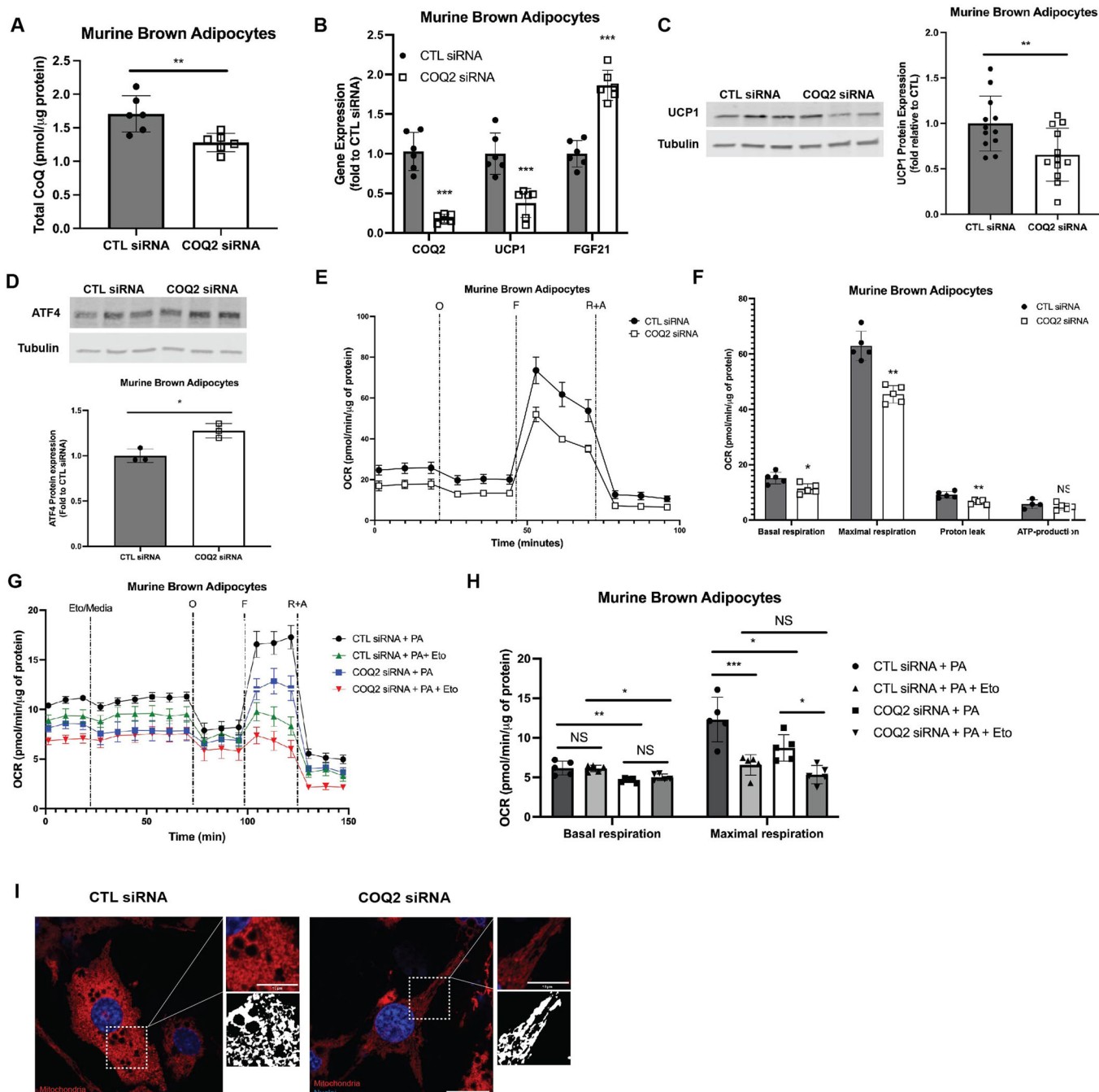

**Figure EV2. Genetically induced CoQ deficiency in murine brown adipocytes via knockdown of COQ2.**

(A) Total CoQ levels, including $CoQ_9$ and $CoQ_{10}$ isoforms, of murine brown adipocytes transfected with either a scramble siRNA (CTL siRNA) or COQ2 siRNA, $n = 6$. (B) Gene expression of CTL siRNA and COQ2 siRNA-treated cells, $n = 5$. (C) UCP1 protein expression of CTL siRNA and COQ2 siRNA-treated cells, $n = 12$. (D) ATF4 protein expression of CTL siRNA and COQ2 siRNA treated cells, $n = 3$. (E) Oxygen consumption rate (OCR) of CTL siRNA and COQ2 siRNA treated cells during mitochondrial stress test, $n = 5$. (F) Basal, maximal, protein leak and ATP production respiration of CTL siRNA and COQ2 siRNA treated cells calculated from oxygen trace shown in (E), $n = 5$. (G) Fatty acid oxidation assay was performed on cells treated with CTL siRNA + palmitate (PA), CTL siRNA + PA + Etomoxir (Eto), COQ2 siRNA + PA and COQ2 siRNA + PA + Eto and OCR was measured, $n = 5$. (H) Basal and maximal respiration of the same cells whose oxygen trace is shown in (E), $n = 5$. Results were compared using a one-way ANOVA test. (I) Brown adipocytes with CTL siRNA or COQ2 siRNA treatment were stained with MitoTracker deep red, and dapi and visualized using the Zeiss LSM880, scale bar = 20 μm. Data information: Data are mean ± SEM. (A–D, F) Results were compared using an unpaired two-tailed Student's $t$ test. Significance presented at *$P < 0.05$, **$P < 0.01$, and ***$P < 0.001$ compared to controls. Data describes biological replicates.

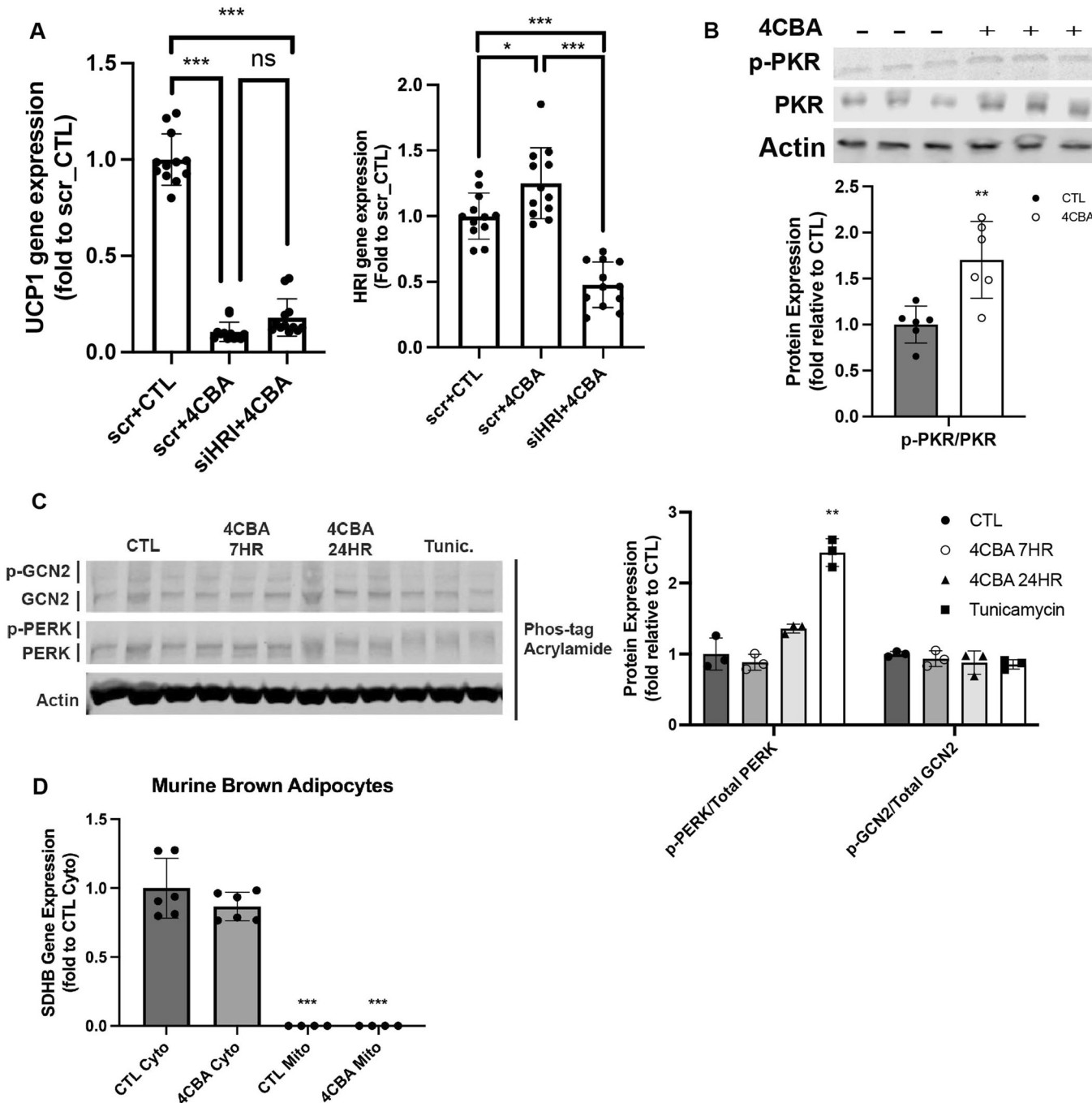

**Figure EV3.  Targeting stress kinases to study ISR induction.**

(**A**) *UCP1* and *HRI* gene expression in brown adipocytes transfected with scramble or HRI siRNA under vehicle control or 4CBA treatment for 24 h, $n = 12$. Pooled data for three repeated experiments. Results were compared using a one-way ANOVA test. (**B**) Western blot analysis of PKR and phosphorylated PKR (Thr451) levels, $n = 6$. (**C**) Phos-tag gel analysis of p-GCN2 and p-PERK compared to total GCN2 and total PERK, $n = 3$. (**D**) SDHB gene expression of cytosolic or mitochondrial cell fractions from vehicle or 4CBA-treated cells, $n = 4$–6. Data information: Data are mean ± SEM. (**B**–**D**) Results were compared using an unpaired two-tailed Student's *t* test. Significance presented at \*$P < 0.05$, \*\*$P < 0.01$, and \*\*\*$P < 0.001$ compared to controls. Data describes biological replicates.

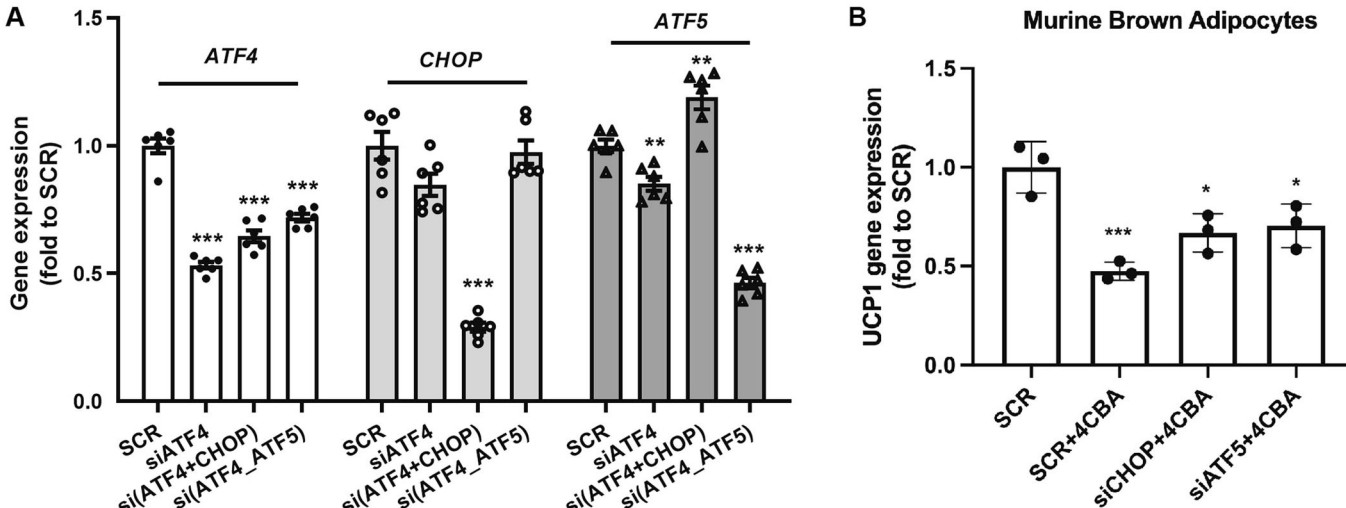

**Figure EV4.  Targeting ISR-related components.**

(A) *ATF4*, *CHOP*, and *ATF5* expression in brown adipocytes with 4CBA treatment after indicated siRNA knockdown of target genes (*x*-axis), $n = 6$. Results were compared using an unpaired two-tailed Student's *t* test. (B) *UCP1* expression in 4CBA-treated brown adipocyte with siRNA-mediated knockdown of *CHOP* or *ATF5*, $n = 3$. Results were compared using a one-way ANOVA test. Data information: Data are mean ± SEM. Significance presented at *$P < 0.05$, **$P < 0.01$, and ***$P < 0.001$ compared to controls. Data describes biological replicates.

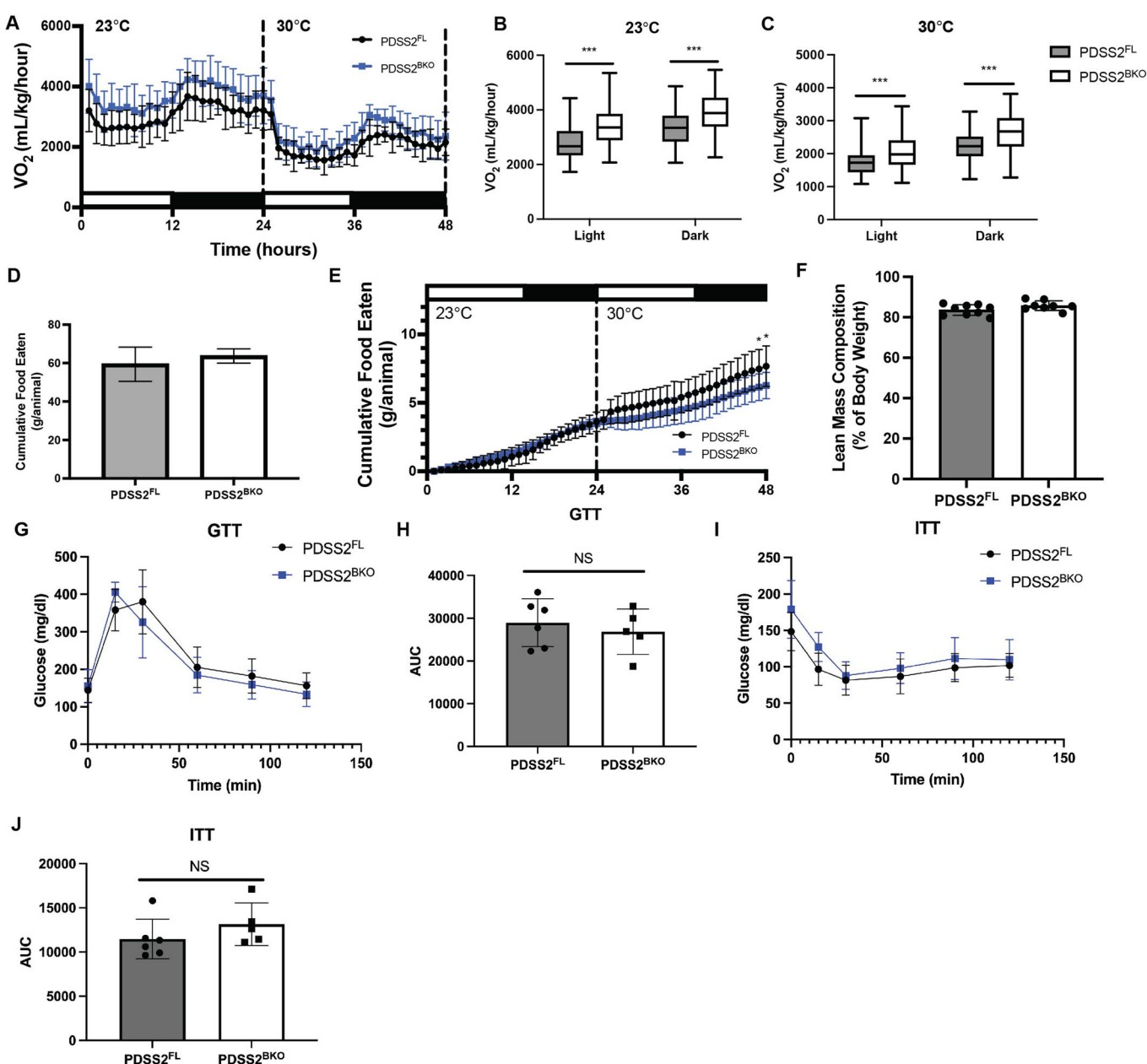

**Figure EV5. Metabolic parameters of PDSS2 knockouts.**

(A) Oxygen consumption rate of PDSS2^FL or PDSS2^BKO animals was monitored under 23 and 30 °C in real-time in metabolic cages using CLAMS. Data normalized to body weight. (B, C) Averaged oxygen consumption rate. Centerline represents median and box extends from 25th to 75th percentiles. Data normalized to body weight. (D) Food consumption accumulated in three weeks. (E) Real-time feeding data from metabolic cage run at 23 and 30 °C. (F) Lean mass composition of PDSS2^FL and PDSS2^BKO animals. (G) Glucose tolerance test (GTT) of PDSS2^FL and PDSS2^BKO animals. 2.0 g/kg body weight glucose dose administered via intraperitoneal (IP) injection. (H) Area under curve (AUC) analysis of GTT presented in (G). (I) Insulin tolerance test (ITT) of PDSS2^FL and PDSS2^BKO animals. 0.4 IU/kg body weight insulin dose administered via IP injection. (J) Area under curve (AUC) analysis of ITT presented in (I). Data information. (A–C) $n = 10$–11/group. (D–F) $n = 8$–9/group. (G–J) $n = 5$–6/group. Data are mean ± SEM. Results were compared using an unpaired two-tailed Student's $t$ test. Significance presented at *$P < 0.05$, and ***$P < 0.001$ compared to controls. Data describes biological replicates.

