## [Peer Review File · The EMBO Journal]

Brown adipose tissue CoQ deficiency activates the integrated stress response and FGF21-dependent mitohormesis

Ching-Fang Chang, Amanda Gunawan, Irene Liparulo, Peter-James Zushin, Kaitlyn Vitangcol, Greg Timblin, Kaoru Sajio, Biao Wang, Güneş Parlakgöl, Ana Arruda, and Andreas Stahl

DOI: [10.15252/emj.2023114056](https://doi.org/10.15252/emj.2023114056)

Corresponding author: *Andreas Stahl (astahl@berkeley.edu)*

Review Timeline:

Submission Date:	17th Mar 23
Editorial Decision:	24th Apr 23
Revision Received:	7th Sep 23
Editorial Decision:	19th Oct 23
Revision Received:	27th Oct 23
Accepted:	8th Nov 23

Editor: *Daniel Klimmeck*

Transaction Report:

Dear Dr Stahl,

Thank you again for the submission of your manuscript (EMBOJ-2023-114056) to The EMBO Journal, as well as for your patience with our feedback at this time of the year. As mentioned, your manuscript has been sent to three referees with expertise in adipocyte biology and metabolic disease, and we have received reports from all of them which are enclosed below.

As you will see from their comments, the referees acknowledge the potential interest and value of your findings on induction of integrated stress response and FGF21-dependent alterations of systemic energy metabolism following depletion of mitochondrial CoQ function in brown adipose tissue. However, they also express major concerns, which need to be fully addressed before they can be supportive of publication of this work at the EMBO Journal. In more detail, referee #1 states substantial issues regarding the depth of insights into CoQ dependent alterations of systemic metabolism (ref#1 pt.a). This expert also requests better exploration of the human relevance of your findings (ref#1 pt.c). Further, reviewer #3 states that the connection between CoQ function and UCP1-independent thermogenic pathways remains too unclear currently and more focus should be put on this aspect (ref#3 pt.1). The referee also states that additional metabolic analyses are required to consolidate the anti-obesity effects reported (ref#3 pt.3; see also ref#1, pt.1).

Given the overall interest stated and broader angle of your findings, we can invite you to revise your manuscript experimentally to address the referees' comments. I need to stress though that we do require strong support from the referees on a revised version of the study to move on to publication of the work. As to the open outcome of the revision work I suggest keeping EMBO Reports in mind for this study as an alternative venue.

In light of the extensive experimentation requested, I would appreciate if you could contact me during the next weeks for exchange e.g. a video call to discuss your perspective on the comments and potential plan for revisions.

When submitting your revised manuscript, please carefully review the instructions below.

Please feel free to approach me any time should you have additional questions related to this.

Thank you for the opportunity to consider your work for publication.

I look forward to your revision.

Kind regards,

Daniel Klimmeck

Daniel Klimmeck, PhD
Senior Editor
The EMBO Journal

Instruction for the preparation of your revised manuscript:

- 1) a .docx formatted version of the manuscript text (including legends for main figures, EV figures and tables). Please make sure that the changes are highlighted to be clearly visible.
- 2) individual production quality figure files as .eps, .tif, .jpg (one file per figure).
- 3) a .docx formatted letter INCLUDING the reviewers' reports and your detailed point-by-point response to their comments. As part of the EMBO Press transparent editorial process, the point-by-point response is part of the Review Process File (RPF), which will be published alongside your paper.
- 4) a complete author checklist, which you can download from our author guidelines (<https://wol-prod-cdn.literatumonline.com/pb->

assets/embo-site/Author Checklist%20-%20EMBO%20J-1561436015657.xlsx). Please insert information in the checklist that is also reflected in the manuscript. The completed author checklist will also be part of the RPF.

6) It is mandatory to include a 'Data Availability' section after the Materials and Methods. Before submitting your revision, primary datasets produced in this study need to be deposited in an appropriate public database, and the accession numbers and database listed under 'Data Availability'. Please remember to provide a reviewer password if the datasets are not yet public (see <https://www.embopress.org/page/journal/14602075/authorguide#datadeposition>).

7) Our journal encourages inclusion of *data citations in the reference list* to directly cite datasets that were re-used and obtained from public databases. Data citations in the article text are distinct from normal bibliographical citations and should directly link to the database records from which the data can be accessed. In the main text, data citations are formatted as follows: "Data ref: Smith et al, 2001" or "Data ref: NCBI Sequence Read Archive PRJNA342805, 2017". In the Reference list, data citations must be labeled with "[DATASET]". A data reference must provide the database name, accession number/identifiers and a resolvable link to the landing page from which the data can be accessed at the end of the reference. Further instructions are available at .

8) At EMBO Press we ask authors to provide source data for the main and EV figures. Our source data coordinator will contact you to discuss which figure panels we would need source data for and will also provide you with helpful tips on how to upload and organize the files.

Numerical data can be provided as individual .xls or .csv files (including a tab describing the data). For 'blots' or microscopy, uncropped images should be submitted (using a zip archive or a single pdf per main figure if multiple images need to be supplied for one panel). Additional information on source data and instruction on how to label the files are available at .

9) We replaced Supplementary Information with Expanded View (EV) Figures and Tables that are collapsible/expandable online (see examples in <https://www.embopress.org/doi/10.15252/emj.201695874>). A maximum of 5 EV Figures can be typeset. EV Figures should be cited as 'Figure EV1, Figure EV2' etc. in the text and their respective legends should be included in the main text after the legends of regular figures.

11) For data quantification: please specify the name of the statistical test used to generate error bars and P values, the number (n) of independent experiments (specify technical or biological replicates) underlying each data point and the test used to calculate p-values in each figure legend. The figure legends should contain a basic description of n, P and the test applied. Graphs must include a description of the bars and the error bars (s.d., s.e.m.).

We realize that it is difficult to revise to a specific deadline. In the interest of protecting the conceptual advance provided by the work, we recommend a revision within 3 months (23rd Jul 2023). Please discuss the revision progress ahead of this time with the editor if you require more time to complete the revisions.

Referee #1:

Summary

The current manuscript by Chang et al. explores the role of Coenzyme Q (CoQ) in brown adipose tissue (BAT) biology. In particular, the authors demonstrate by RNA Seq analysis in mature brown adipocytes treated with the CoQ biosynthesis inhibitor 4CBA that CoQ inhibition significantly changed genes involved in mitochondrial function, integrated stress response (ISR) and the mitochondrial unfolded protein response (UPR), validated by studies in animals specifically lacking PDSS2, a key enzyme in CoQ biosynthesis, in BAT. Upon loss of CoQ, brown adipocytes showed an altered mitochondrial morphology and an accumulation of mitochondrial RNA. Furthermore, 4CBA-mediated suppression of UCP1 expression was partially rescued by the ISR component kinase PKR. In line with these findings, suppression of UCP1 expression by 4CBA exposure could also be rescued by the ISR downstream target, transcription factor ATF4. PDSS2 BAT-specific KO mice displayed enhanced oxygen consumption rates, despite lower UCP1 expression, correlating with reduced locomotor activity and no alteration in food intake, however increased glucose usage. These metabolic phenotypes went along with an increase in BAT FGF21 expression. In BAT-specific PDSS2/FGF21 double KO animals, circulating FGF21 levels were normalized and oxygen consumption rates were lower as compared with PDSS2 KO alone. Finally, UCP1-independent thermogenic pathways, including the phospho-creatine cycle, the calcium cycle and proton leak, were changed in extra BAT organs, e.g. liver, kidney, muscle and white adipose tissue (WAT). Overall, the authors conclude that these results demonstrate a novel function of CoQ in whole-body energy metabolism with far-ranging implications for states of CoQ deficiencies.

General comments

Mitochondrial dysfunction is a key event in several severe human pathologies, including obesity, diabetes, and NASH. Therefore, advancing our understanding of mitochondrial function and dysfunction is still a key challenge in biomedical research. By studying the metabolic consequences of CoQ deficiency, the current manuscript by Chang et al. thereby contributes to a timely and important topic in biomedical research. The authors employ state-of-the-art technology with a wide range of cellular and animal models to study CoQ biology. The manuscript is well-written, concise and well-structured. In general, the conclusions are supported by the experimental data. However, three main issues require additional attention by the authors: a) The underlying basis for the observed metabolic phenotype in PDSS2 BAT KO mice has not really been resolved yet. There is no clear demonstration that indeed extra BAT organs compensate for the loss of UCP1 in BAT, particularly given the fact that e.g. the phospho-creatine cycle is supposed to be BAT-specific and non-functional in other organs. In the absence of changes in food intake and even a reduction in physical activity, the oxygen must be produced somewhere. Also, the increase in RER suggests that the animals preferentially burn glucose. Additional parameters of glucose metabolism should be measured, including e.g. GTTs etc. b) Many in vitro data rely on the use of the pharmacological inhibitor 4CBA. How do you verify the effectiveness of the treatment across experiments? Key findings in vitro should be supported by genetically-induced CoQ deficiency, e.g. by employing siRNA against key biosynthesis enzymes and then demonstrating similar phenotypes. c) It remains unclear to which degree the observed findings relate to human pathophysiology. Do you observe activation of the ATF4 pathway in human primary or secondary CoQ deficiency? Is the CoQ pathway altered in fatty liver disease/NASH etc? Validation of key findings in human cells -at least- and adding discussion to the relevant disease conditions seems appropriate. The addition of corresponding data will significantly strengthen the case for publication.

Specific comments

Fig. 1/3: Please provide evidence for the effectiveness and specificity of the pharmacological treatment. Also, verify pharmacological approaches by siRNA etc genetic perturbation of the CoQ pathway.

Fig. 4i: Please show all data including the control group set as 1 and then all other groups in relation to the control. Do not show degree of rescue.

Fig. 5: Please provide data on fecal energy content to complete the overall assessment of energy balance.

Fig. 5: How do these animals respond to a 4 degree cold challenge? Are they cold intolerant?

Fig. 6: Panel labels in text do not fit the actual figure composition. Please correct.

Fig. 6 E-H: Do the single and double KO mice have the same floxed controls? If not, please show all four groups.

Fig. 6 G-H: Please also include floxed controls in these panels and show the entire oxygen traces as in 5A.

Fig. 6: Is oxygen consumption in isolated mitochondria from single and DKO mice altered? This could clarify whether the energy costly processes triggered by BAT CoQ deficiency indeed reside within extra BAT organs.

Fig. 7: Please show gene expression levels relative to BAT levels.

Referee #2:

The manuscript by Chang and colleagues describes a connection between CoQ deficiency, the integrated stress response, and FGF21 in brown adipose tissue that affects whole body energy expenditure. This is an interesting study that delves into a poorly understood aspect of CoQ biology and uncovers findings related to CoQ function and brown fat that are of broad interest. However, there are several typos in the manuscript, misplaced figures, and at least in one case a duplicated Western blot. These need to be corrected in a very thorough proofreading before this manuscript can be fully considered for an initial review. A few of additional concerns are listed below.

1. There are several issues with blots, figures, and panels etc. that need to address before moving forward with the initial review. For example, Figure 3D and Figure S2C appear to show the same actin blot, the latter not matching with the number of lanes shown above it. The authors need to do a careful proofreading before this is ready for consideration as the manuscript seems rushed. Some other points to check:

Switched blots:

- p-PKR and PKR Supplemental Figure 2B.

Unclear methods:

- Was 4CBA drug diluted in DMSO or ethanol? vehicle states to be DMSO or Ethanol.

- Can authors please add drug dosage in Methods for human beige adipocytes Fig1H, or clarify it is the same.

Microscopy:

- ensure scale bars and labeling are the same Figure 2A, address changes in lipid droplet accumulation.

Mislabeled Figure Panels/Text:

- Figure 3 has two Ds.

- In paragraph ATF4-FGF21, FGF21 transcript increase, should be Figure 4E and not D.

- In text, states that Fig6C-D show body and adipose tissue weight, mislabeled, please correct. Fig6E-F do not show VO2 either please correct accordingly.

- Figure 3D (should be 3C), it is unclear that p-PKR remains high at 24h, please enlarge this blot for better visualization.

Formatting:

- In Introduction can you please expand abbreviation (UPRmt).

- Induction of ISR.. paragraph, "after" is doubled on line 2.

- Please ensure that all formatting is the same, e.g Figure 6 D blots, and please include tissue label that it is Liver, as in iWAT blot.

2. The authors generate their hypothesis in part from in vitro RNA-sequencing data that uses a COQ2 inhibitor to block CoQ biosynthesis. Please address to what extent these changes observed are unique to brown adipocytes versus a general cellular response to a deficiency in CoQ. Also, the concentration of 4CBA is around 4mM, which seems high. Please describe how these concentrations were arrived at and whether a titration or genetic model should be considered.

3. How do the authors account for the finding that UCP1 gene expression is significantly decreased by 50% after only 2 hours of 4CBA treatment, but CoQ only drops to 75% after 7 hours of treatment? Does this effect on UCP1 gene expression precede a drop in CoQ levels? If yes, how?

4. The authors describe the use of Phos-tag gels to detect phosphorylation changes on PKR, this is difficult to see on the shown images as there appears to be similar lettering in every lane. In the same figure, the author's show differential accumulation of mitochondrial transcripts in different cellular fractions, but do not show markers of successful fractionation.

5. How FGF21 might be acting in this context (e.g. target tissue) has not been sufficiently resolved and whether BAT specifically is important vs other tissues is not well shown (low N Figure 6C).

6. The sex of the mice used needs to be clearly indicated and commented on in the manuscript.

7. Validation of KOs should be shown (e.g PDSS2 KO F1G etc.)

Referee #3:

In the manuscript titled "Brown adipose tissue CoQ deficiency activates the integrated stress response and FGF21 dependent mitohormesis" by Chang et al., the authors propose a signaling pathway through Coenzyme Q (CoQ) deficiency in brown fat, activation of integrated stress response (ISR) and FGF21 mediated increased energy expenditure. The concept is novel and potentially interesting. The authors use multiple genetic mouse models to investigate this signaling, including CoQ synthesis enzyme decaprenyl diphosphate synthase subunit 2 (PDSS2) flox mice crossed with UCP1^{cre} to generate PDSS2-BKO mice, FGF21 flox mice and ATF4 flox mice crossed with PDSS2 BKO to generate DKO mice for either FGF21 or ATF4. The observation that BAT CoQ deficiency caused mitochondrial RNA accumulation which renders ISR through PKR is interesting. Despite of the thought-provoking hypothesis, there are some aspects of the current study should be further addressed before the conclusion can be fully substantiated.

Major points,

1. In Figure 7, the authors included qPCR results of some of the genes on the three known UCP1-independent thermogenic pathways, namely, Serca-Ca²⁺ futile cycle, creatine-dependent pathway and AAC2 mediated proton leak. In addition to three adipose depots, the authors also tested liver, heart, muscle and kidney. It was pointed out that AAC2 is upregulated in all peripheral tissues and SERCA2 and creatine related genes are upregulated in the liver. The authors cited a couple of papers (Ref #52 and Ref #53) as evidence that FGF21 has been implicated in other tissue (in addition to adipose) mediated thermogenesis in previous studies. In both previous studies, comprehensive functional investigation was carried out to characterize how systemic energy expenditure was affected by FGF21, and in both cases, it has been proposed that central response (nervous system) may be responsible for the adipose-independent regulation through FGF21. As of right now, figure 7 is the only provided link between in vivo phenotype (lower body weight, increased energy expenditure) observed in PDSS2-BKO mice and drastic suppressed expression of UCP1. This is a very important point, the authors need to provide more experimental evidence than qPCR results in various tissues. For the results they do provide, some of them were hard to reconcile with what was known about these pathways. for example, there have been significant follow up studies about these UCP1-independent thermogenic pathways (e.g. PMID36344764). The additional regulatory components of these UCP1-independent thermogenic pathways are necessary for the thermogenesis to be functional and have only been characterized and studied in adipocytes, if the authors are proposing that these pathways are working in other peripheral organs, they need to test all these components (such as ADRA1A) and show they are present and functional in these tissues, such as the liver.

2. In figure 5, the authors shown that PDSS2BKO mice have less activity, but did not explore the reason for this. They only concluded that this indicates that low body weight gain on HFD was not due to more activity in BKO mice. Is the reduced activity in PDSS2BKO mice caused by some direct or indirect effects of FGF21 in the nervous system?(as suggested in ref #52 and #53, FGF21 does affect systemic metabolism through the nervous system)

3. Even though the anti-obesity effects were nicely demonstrated in the PDSS2bko mice, the authors did not include any direct metabolic investigation, such as glucose tolerance test or insulin tolerance test. Given the authors were proposing this pathway has mitohormetic effects, then actually demonstrate the metabolic benefits, not just weight loss, would be necessary.

4. In Figure 2, the authors included results showing how mitochondrial fission and Drp1 phosphorylation at serine 616 (previously shown to be PKA dependent) are affected by CoQ2 inhibitor 4CBA. This information is not well integrated to the whole paper as of right now and seems out of place.

Minor points

1. There were some typos within the manuscript and sometimes it did make reading it a bit challenge. For example, ref #70 is basically an empty entry with only the first author partial name. Some additional proof reading would be helpful.

Referee #1:

Summary

The current manuscript by Chang et al. explores the role of Coenzyme Q (CoQ) in brown adipose tissue (BAT) biology. In particular, the authors demonstrate by RNA Seq analysis in mature brown adipocytes treated with the CoQ biosynthesis inhibitor 4CBA that CoQ inhibition significantly changed genes involved in mitochondrial function, integrated stress response (ISR) and the mitochondrial unfolded protein response (UPR), validated by studies in animals specifically lacking PDSS2, a key enzyme in CoQ biosynthesis, in BAT. Upon loss of CoQ, brown adipocytes showed an altered mitochondrial morphology and an accumulation of mitochondrial RNA. Furthermore, 4CBA-mediated suppression of UCP1 expression was partially rescued by the ISR component kinase PKR. In line with these findings, suppression of UCP1 expression by 4CBA exposure could also be rescued by the ISR downstream target, transcription factor ATF4. PDSS2 BAT-specific KO mice displayed enhanced oxygen consumption rates, despite lower UCP1 expression, correlating with reduced locomotor activity and no alteration in food intake, however increased glucose usage. These metabolic phenotypes went along with an increase in BAT FGF21 expression. In BAT-specific PDSS2/FGF21 double KO animals, circulating FGF21 levels were normalized and oxygen consumption rates were lower as compared with PDSS2 KO alone. Finally, UCP1-independent thermogenic pathways, including the phospho-creatine cycle, the calcium cycle and proton leak, were changed in extra BAT organs, e.g. liver, kidney, muscle and white adipose tissue (WAT). Overall, the authors conclude that these results demonstrate a novel function of CoQ in whole-body energy metabolism with far-ranging implications for states of CoQ deficiencies.

General comments

Mitochondrial dysfunction is a key event in several severe human pathologies, including obesity, diabetes, and NASH. Therefore, advancing our understanding of mitochondrial function and dysfunction is still a key challenge in biomedical research. By studying the metabolic consequences of CoQ deficiency, the current manuscript by Chang et al. thereby contributes to a timely and important topic in biomedical research. The authors employ state-of-the-art technology with a wide range of cellular and animal models to study CoQ biology. The manuscript is well-written, concise and well-structured. In general, the conclusions are supported by the experimental data. However, three main issues require additional attention by the authors: a) The underlying basis for the observed metabolic phenotype in PDSS2 BAT KO mice has not really been resolved yet. There is no clear demonstration that indeed extra BAT organs compensate for the loss of UCP1 in BAT, particularly given the fact that e.g. the phospho-creatine cycle is supposed to be BAT-specific and non-functional in other organs. In the absence of changes in food intake and even a reduction in physical activity, the oxygen must be produced somewhere. Also, the increase in RER suggests that the animals preferentially burn

glucose. Additional parameters of glucose metabolism should be measured, including e.g. GTTs etc. b) Many in vitro data rely on the use of the pharmacological inhibitor 4CBA. How do you verify the effectiveness of the treatment across experiments? Key findings in vitro should be supported by genetically-induced CoQ deficiency, e.g. by employing siRNA against key biosynthesis enzymes and then demonstrating similar phenotypes. c) It remains unclear to which degree the observed findings relate to human pathophysiology. Do you observe activation of the ATF4 pathway in human primary or secondary CoQ deficiency? Is the CoQ pathway altered in fatty liver disease/NASH etc? Validation of key findings in human cells -at least- and adding discussion to the relevant disease conditions seems appropriate.

a)

- i) We have done extensive additional experiments to address this point and to further define the mitohormetic response following BAT CoQ deficiency. To understand if organs outside of BAT are contributing to the enhanced respiration we observe in PDSS2^{BKO} animals we measured oxygen consumption in inguinal white adipose tissue (iWAT) tissue chunks as well as in primary hepatocytes isolated from PDSS2^{FL} and PDSS2^{BKO} animals. We chose these two organs since adipose and liver are both known targets of FGF21's effects and we demonstrated that the compensatory effects require FGF21. We found that iWAT had enhanced oxygen consumption in the PDSS2^{BKO} animals (Fig. 7A-B) and dramatically increased being as observed by the presence of more multilocular adipocytes with smaller lipid droplets (Fig. 7G) and increased UCP1 expression (Fig. 7E, G). Since there were no significant changes to iWAT CoQ levels (Fig. 7F), we concluded that these changes were due to a signal coming from dysfunctional BAT. Unlike iWAT, primary hepatocytes from PDSS2^{BKO} animals did not exhibit increased oxygen consumption (Fig. EV7E-F). We also confirmed that BAT was not contributing to the enhanced respiration since BAT tissue pieces from the PDSS2^{BKO} animals had mildly lower respiration than tissue pieces from PDSS2^{FL} (Fig. EV7A-B).
- ii) In order to further study glucose metabolism in the model, we performed GTT and ITT in PDSS2^{BKO} animals to see if insulin action is altered. We did these measurements using animals fed a defined low CoQ diet for 2 weeks. We did not find any significant differences in insulin or glucose tolerance when comparing the PDSS2^{FL} and PDSS2^{BKO} mice (Fig. EV5G-J). To look at other facets of glucose metabolism we measured markers of gluconeogenesis in liver of the PDSS2^{BKO} animals fed a high fat diet and found that gene expression of glucose-6-phosphatase and PEPCK was increased (Fig. 7J). This further supports FGF21's mitohormetic effects on the liver in the face of BAT CoQ deficiency since FGF21 is known to increase hepatic gluconeogenesis (Wang, C. et al., 2014). Further, Glut1, a known downstream mediator of FGF21's effects (Ge, X. et al., 2011), was

observed to have increased expression in tissues of PDSS2^{BKO} animals such as liver, eWAT and muscle (Fig. 7I).

- b) We experimentally addressed this point and have verified that 4CBA treatment leads to decreased CoQ levels in cell culture in a time-dependent manner (Fig. 4A) and, from previous findings, we also observed that PDSS2^{BKO} animals have lower CoQ levels in BAT (Chang, C. F. et al., 2022). We will cite this in the manuscript. In addition to the PDSS2^{BKO} genetic animal model that we already have, we used a COQ2 siRNA to induce knockdown in our immortalized murine brown adipocyte cell line. We found that COQ2 knockdown phenocopied 4CBA effects and resulted in lower CoQ levels, decreased expression in UCP1, and induction of the stress response as demonstrated through increase in FGF21 gene and ATF4 protein expression (Fig. EV2 A-D). Further we found that, in line with lower UCP1 expression, COQ2 knockdown cells had lower cellular respiration when exposed to a mitochondrial stress test and a fatty acid oxidation assay (Fig. EV2 E-H), we had previously reported that 4CBA treatment similarly causes decreased cellular respiration (Chang, C. F. et al., 2022).
- c) We show in Fig. 1H that human beige adipocytes treated with 4CBA have lower UCP1 gene expression and increased gene expression of factors involved in the UPR^{mt} and ISR. Some comparisons can be made between literature about relevant patient data and the model that we propose. Decreased OXPHOS respiration and ROS production are common aspects to several diseases linked to CoQ deficiency (Quinzii, C. M. & Hirano, M., 2010), both of which we see in the BAT-CoQ deficiency model (Chang, C. F. et al., 2022). Also, it has been reported that FGF21 is a potential valid marker for mitochondrial disorders such as OXPHOS disease in patients (Suomalainen, A. et al., 2011). Regarding NAFLD, another study has shown that CoQ10 supplementation attenuates high-fat-diet induced NAFLD in mice through the activation of the AMPK pathway (Chen, K. et al., 2019).

The addition of corresponding data will significantly strengthen the case for publication.

Specific comments

Fig. 1/3: Please provide evidence for the effectiveness and specificity of the pharmacological treatment. Also, verify pharmacological approaches by siRNA etc genetic perturbation of the CoQ pathway.

As mentioned above we show in Fig. 4A that 4CBA treatment successfully lowers CoQ levels in cells. We also showed that COQ2 knockdown exhibits similar phenotypes to 4CBA treatment in Fig. EV2.

Fig. 4i: Please show all data including the control group set as 1 and then all other groups in relation to the control. Do not show degree of rescue.

We edited this figure so that it shows fold to the control group instead of percent of rescue.

Fig. 5: Please provide data on fecal energy content to complete the overall assessment of energy balance.

Given the FGF21 dependence of the enhanced respiration phenotype, we do not think that these data would be critical to the interpretation of the manuscript and, in the absence of a clear hypothesis involving malabsorption, we cannot justify the considerable investment into testing this (please note that UC Berkeley has no facilities to support fecal calorimetry).

Fig. 5: How do these animals respond to a 4 degree cold challenge? Are they cold intolerant?

Yes, these animals are indeed cold intolerant. When we put them in a 4-degree cold challenge their oxygen consumption starts to plummet and we have to take them out of the cold for animal welfare concerns after just 6 hours. We published this data in a previous publication (Chang, C. F. et al., 2022). We also cited this in our revised manuscript.

Fig. 6: Panel labels in text do not fit the actual figure composition. Please correct. We edited the panel labels so that they fit with the figure composition. We apologize for these typos.

Fig. 6 E-H: Do the single and double KO mice have the same floxed controls? If not, please show all four groups.

For Fig. 6F-G, formerly Fig. 6E-F, the single and double KO mice have the same floxed controls and they are now shown in the figure.

Fig. 6 G-H: Please also include floxed controls in these panels and show the entire oxygen traces as in 5A.

For Fig. 6I-J, formerly Fig. 6G-H, we added the floxed controls to these figures and we also included the entire oxygen traces in Fig. 6H.

Fig. 6: Is oxygen consumption in isolated mitochondria from single and DKO mice altered? This could clarify whether the energy costly processes triggered by BAT CoQ deficiency indeed reside within extra BAT organs.

As mentioned above we measured oxygen consumption in BAT tissue pieces and found that there was a mild decrease in oxygen consumption (Fig. EV7A-B), while iWAT tissue pieces had an increase in oxygen consumption (Fig. 7C-D) showing that indeed peripheral organs contribute to the observed enhanced energy expenditure.

Fig. 7: Please show gene expression levels relative to BAT levels.

We have heavily rearranged Fig. 7 so that now the UCP1-independent thermogenic gene expression analysis is shown in Fig. 7H, EV7D and EV7G-K. With this new formatting it doesn't make as much sense to show gene expression relative to BAT levels, since Fig. 7 focuses on the iWAT phenotype.

Referee #2:

The manuscript by Chang and colleagues describes a connection between CoQ deficiency, the integrated stress response, and FGF21 in brown adipose tissue that affects whole body energy expenditure. This is an interesting study that delves into

a poorly understood aspect of CoQ biology and uncovers findings related to CoQ function and brown fat that are of broad interest. However, there are several typos in the manuscript, misplaced figures, and at least in one case a duplicated Western blot. These need to be corrected in a very thorough proofreading before this manuscript can be fully considered for an initial review. A few of additional concerns are listed below.

1. There are several issues with blots, figures, and panels etc. that need to address before moving forward with the initial review. For example, Figure 3D and Figure S2C appear to show the same actin blot, the latter not matching with the number of lanes shown above it. The authors need to do a careful proofreading before this is ready for consideration as the manuscript seems rushed. Some other points to check:

The figure does not show a duplicated western blot but does contain an error we are thankful to the reviewer for pointing out. There is a mistake with the cropping of the actin blot, shown below. We've fixed this. We also provided all uncropped western blots in the source data folder of our resubmission.

Switched blots:

- p-PKR and PKR Supplemental Figure 2B.

These blots are not switched. We are unsure what the reviewer is referring to.

Unclear methods:

- Was 4CBA drug diluted in DMSO or ethanol? vehicle states to be DMSO or Ethanol.

4CBA was diluted in a mixture of DMSO and ethanol as stated in the manuscript.

- Can authors please add drug dosage in Methods for human beige adipocytes Fig1H, or clarify it is the same.

The drug dosage for the human beige adipocytes is the same as the murine brown adipocytes, 4mM. We will clarify this in the methods section.

Microscopy:

- ensure scale bars and labeling are the same Figure 2A, address changes in lipid droplet accumulation.

The scale bar is visible. If we need to enlarge or add it for the magnification of mitochondria details we can. We added comments on how visually it looks like there are larger lipid droplets in 4CBA treated murine brown adipocytes in our discussion section. This is in line with the larger lipid droplet morphology we observe in H&E histology of BAT from PDSS2^{BKO} animals that we reported in on in a previous publication (Chang et al., 2022). Moreover, it is known that CoQ is an essential cofactor for beta fatty oxidation (Crane et al. 2013), which we show is decreased in COQ2 knockdown cells via a Palmitate Oxidation Stress Test (Fig. EV2G-H). We could hypothesis that a decreased utilization of lipids could lead to an accumulation of free fatty acids (FFAs) and an increase in lipid droplet size.

Mislabeled Figure Panels/Text:

- Figure 3 has two Ds.

We apologize for this typo and have fixed it.

- In paragraph ATF4-FGF21, FGF21 transcript increase, should be Figure 4E and not D.

We apologize for this typo and we have fixed it.

- In text, states that Fig6C-D show body and adipose tissue weight, mislabeled, please correct. Fig6E-F do not show VO2 either please correct accordingly.

We apologize for this typo and we have fixed it.

- Figure 3D (should be 3C), it is unclear that p-PKR remains high at 24h, please enlarge this blot for better visualization.

We increased the size of this blot for better visualization.

Formatting:

- In Introduction can you please expand abbreviation (UPRmt).

We expanded this abbreviation

- Induction of ISR.. paragraph, "after" is doubled on line 2.

We fixed this typo.

- Please ensure that all formatting is the same, e.g Figure 6 D blots, and please include tissue label that it is Liver, as in iWAT blot.

We formatted these western blots correctly and also added labels.

2. The authors generate their hypothesis in part from in vitro RNA-sequencing data that uses a COQ2 inhibitor to block CoQ biosynthesis. Please address to what

extent these changes observed are unique to brown adipocytes versus a general cellular response to a deficiency in CoQ. Also, the concentration of 4CBA is around 4mM, which seems high. Please describe how these concentrations were arrived at and whether a titration or genetic model should be considered.

In previous studies we showed that CoQ deficiency does not induce changes in respiration or UCP1 expression in 3T3L1 white adipocytes (Chang, C. F. et al, 2022). We will cite this in the manuscript. Further, in this manuscript we also show that when we use a liver specific (albumin-cre) knockout of PDSS2 in mice we do not see an induction in the integrated stress response (Fig. EV6A). From these data we show that these findings are unique to brown/beige adipose tissue.

We based the 4-CBA treatment concentration according to previous literature using this inhibitor to achieve CoQ deficiency. (Bersuker et al., 2019, Koppula et al., 2022, Deshwal et al, 2023). These studies have demonstrated that millimolar rather than micromolar concentrations of 4CBA, are effective in inducing CoQ deficiency in different types of cells and experimental models, showing no off-target effects.

3. How do the authors account for the finding that UCP1 gene expression is significantly decreased by 50% after only 2 hours of 4CBA treatment, but CoQ only drops to 75% after 7 hours of treatment? Does this effect on UCP1 gene expression precede a drop in CoQ levels? If yes, how?

We measured CoQ levels following 2 hours of 4CBA treatment and found it to be decreased to 75% of control cells (Fig 4A).

4. The authors describe the use of Phos-tag gels to detect phosphorylation changes on PKR, this is difficult to see on the shown images as there appears to be similar lettering in every lane. In the same figure, the author's show differential accumulation of mitochondrial transcripts in different cellular fractions, but do not show markers of successful fractionation.

- This is a quantitative not qualitative effect. The number of bands does not necessarily change between the control and treated samples, but one can clearly see that after 7 hours of treatment the uppermost band is significantly stronger compared to the control. This band represents the most phosphorylated form of PKR, showing that phosphorylation status of PKR changes. We also quantified this change as seen in the figure. We have enlarged this western blot so this change can be seen more clearly.

- We thank the reviewer for their suggestion about cellular fractionation. We measured gene expression of SDHB in the cytosolic and mitochondrial fraction and found that expression was enriched in the cytosol but not the mitochondria (Fig. EV3D). This shows successful fractionation since SDHB is a mitochondrially localized protein that is nuclear encoded so there should be no presence of its transcripts in the mitochondria.

5. How FGF21 might be acting in this context (e.g. target tissue) has not been sufficiently resolved and whether BAT specifically is important vs other tissues is not well shown (low N Figure 6C).

- In order to show that peripheral organs are in fact contributing to enhanced whole body respiration in the PDSS2^{BKO} mice we measured oxygen consumption in iWAT tissue pieces and primary hepatocytes. We found that iWAT had enhanced oxygen consumption in the PDSS2^{BKO} animals (Fig. 7A-B) and increased beigeing as observed by the presence of more multilocular adipocytes with smaller lipid droplets (Fig. 7G) and increased UCP1 expression (Fig. 7E, G). Since there were no significant changes to iWAT CoQ levels (Fig. F), we concluded that these changes were due to a signal coming from dysfunctional BAT. Unlike iWAT, primary hepatocytes from PDSS2^{BKO} animals did not exhibit increased oxygen consumption (Fig. EV7E-F). We also confirmed that BAT was not contributing to the enhanced respiration since BAT tissue pieces from the PDSS2^{BKO} animals had mildly lower respiration than chunks from PDSS2^{FL} (Fig. EV7A-B). We also found that known FGF21 regulators were changed in our PDSS2^{BKO} animals. For example, PDSS2^{BKO} animals fed a high fat diet have increases in hepatic gluconeogenic genes, PEPCK and glucose-6-phosphotase (Fig. 7J) and Glut1 expression, an FGF21 mediator (Ge, X. et al., 2011), is increased in extra-BAT tissue in the PDSS2^{BKO} animals including eWAT, liver and muscle (Fig. 7I).
- For figure 6C we use an N=4. For gene expression analysis we believe this is common and standard deviations are not excessive.

6. The sex of the mice used needs to be clearly indicated and commented on in the manuscript.

We use male mice for all experiments in this paper. This is stated in the methods section of the manuscript.

7. Validation of KOs should be shown (e.g PDSS2 KO F1G etc.)

We had previously shown that the PDSS2^{BKO} mice have significant decrease in PDSS2 expression as well as a decrease in BAT CoQ levels (Chang et al., 2022). In the revised manuscript we have included validation of the single and double KO's using gene expression analysis (Fig. EV5A, 6A and EV6B).

Referee #3:

In the manuscript titled "Brown adipose tissue CoQ deficiency activates the integrated stress response and FGF21 dependent mitohormesis" by Chang et al., the authors propose a signaling pathway through Coenzyme Q (CoQ) deficiency in brown fat, activation of integrated stress response (ISR) and FGF21 mediated increased energy expenditure. The concept is novel and potentially interesting. The authors use multiple genetic mouse models to investigate this signaling, including CoQ synthesis enzyme decaprenyl diphosphate synthase subunit 2 (PDSS2) flox mice crossed with UCP1cre to generate PDSS2-BKO mice, FGF21 flox mice and ATF4 flox mice crossed with PDSS2 BKO to generate DKO mice for either FGF21 or ATF4. The observation that BAT CoQ deficiency caused mitochondrial RNA accumulation which renders ISR through PKR is interesting. Despite of the thought-provoking hypothesis, there are some aspects of the current study should be further addressed before the conclusion can be fully substantiated.

Major points,

1. In Figure 7, the authors included qPCR results of some of the genes on the three known UCP1-independent thermogenic pathways, namely, Serca-Ca²⁺ futile cycle, creatine-dependent pathway and AAC2 mediated proton leak. In addition to three adipose depots, the authors also tested liver, heart, muscle and kidney. It was pointed out that AAC2 is upregulated in all peripheral tissues and SERCA2 and creatine related genes are upregulated in the liver. The authors cited a couple of papers (Ref #52 and Ref #53) as evidence that FGF21 has been implicated in other tissue (in addition to adipose) mediated thermogenesis in previous studies. In both previous studies, comprehensive functional investigation was carried out to characterize how systemic energy expenditure was affected by FGF21, and in both cases, it has been proposed that central response (nervous system) may be responsible for the adipose-independent regulation through FGF21. As of right now, figure 7 is the only provided link between in vivo phenotype (lower body weight, increased energy expenditure) observed in PDSS2-BKO mice and drastic

suppressed expression of UCP1. This is a very important point, the authors need to provide more experimental evidence than qPCR results in various tissues. For the results they do provide, some of them were hard to reconcile with what was known about these pathways. For example, there have been significant follow up studies about these UCP1-independent thermogenic pathways (e.g. PMID36344764). The additional regulatory components of these UCP1-independent thermogenic pathways are necessary for the thermogenesis to be functional and have only been characterized and studied in adipocytes, if the authors are proposing that these pathways are working in other peripheral organs, they need to test all these components (such as ADRA1A) and show they are present and functional in these tissues, such as the liver.

In order to address if BAT or other organs are contributing to the enhanced respiration we performed a SeaHorse cellular respiration assay on inguinal white adipose tissue (iWAT) tissue pieces and primary hepatocytes isolated from PDSS2^{BKO} animals. We found that while iWAT from PDSS2^{BKO} had enhanced respiration (Fig. 7A-B), primary hepatocytes did not have significant changes in respiration between floxed controls and knockout animals (Fig. EV7E-F). In addition to enhanced respiration, we also found increased beiging in PDSS2^{BKO} iWAT as observed by the presence of more multilocular adipocytes with smaller lipid droplets (Fig. 7G) and increased UCP1 expression (Fig. 7E, G). We also confirmed that BAT was not contributing to the enhanced respiration since BAT tissue pieces from the PDSS2^{BKO} animals had mildly lower respiration than chunks from PDSS2^{FL} (Fig. EV7A-B). In our discussion we include comments about how increased beiging in iWAT could be due to either a direct effect of FGF21 or an indirect effect of FGF21 acting on the CNS. Further, both BAT and iWAT from PDSS2^{BKO} animals had increases in ECAR rate (Fig. 7C and EV7C), this is in line with the increases in expression of UCP1-independent thermogenic genes that we observe in both BAT and iWAT as this increase in ATP synthesis could help to fuel the futile cycling processes that are increased in these tissues.

We agree that the UCP1 independent thermogenic pathways in non-BAT tissues are of interest. Both the signals from BAT as well as the specific nature of futile cycles in the downstream tissues are subject to significant research efforts by many groups but have remained incompletely understood. Much of the literature on this topic has focused on FGF21 dependent signals. Importantly, we show that in the PDSS2/FGF21 double KO enhanced respiration is no longer observed indicating that much of the effect is via this BAT secreted hormone. We have changed the discussion to include this lack of information in the field in regard to UCP1-independent thermogenic pathways and have included discussion of the potential pathway leading to the beiging we observe in iWAT.

We thank the reviewer for their insightful comments about increased energy expenditure. Further analysis of the increased energy expenditure phenotype and the mitohormesis in the PDSS2^{BKO} mice is indeed an interesting topic for future research in the lab. We have already started to plan experiments to profile the secreted factors coming out of CoQ deficient BAT, which may be leading to

mitohormesis. We will do this using a TurboID system both in vitro and in vivo, which biotinylate proteins secreted from cells/specific tissue types (Wei et al., 2021). This will allow us to identify if FGF21 is the only required factor leading to increased energy expenditure in the model we propose. We will also use co-culture experiments in vitro to further profile the mechanism by which BAT affects other tissues. Overall, in this manuscript, we provide a long chain of mechanistic insight ranging from the initial CoQ defect at the mitochondrial level, the induction of UPRmt, the induction of the ISR via PKR, the ATF4 dependent suppression of UCP1 and concomitant upregulation of FGF21, and the subsequent increased energy expenditure phenotype in BAT CoQ deficient animals potentially stemming from increased beiging in iWAT. However, we believe that extensive investigations into FGF21's specific direct or indirect mechanism of action is beyond the scope of this paper, we will highlight these future directions of study in the discussion of the manuscript.

2. In figure 5, the authors shown that PDSS2BKO mice have less activity, but did not explore the reason for this. They only concluded that this indicates that low body weight gain on HFD was not due to more activity in BKO mice. Is the reduced activity in PDSS2BKO mice caused by some direct or indirect effects of FGF21 in the nervous system?(as suggested in ref #52 and #53, FGF21 does affect systemic metabolism through the nervous system)

We agree that there is a possibility that FGF21 action on the CNS could be involved in the model we propose, we see hints of this from the lower activity (Fig. 5D) as well as lower food intake at thermoneutrality (Fig. EV5F) in the PDSS2^{BKO} mice. We added discussion about potential links to the CNS to the discussion section, but since this is still a field with many unanswered questions we feel as though an in-depth study of the CNS is beyond the scope of this manuscript and believe it would be a topic for future studies.

3. Even though the anti-obesity effects were nicely demonstrated in the PDSS2bko mice, the authors did not include any direct metabolic investigation, such as glucose tolerance test or insulin tolerance test. Given the authors were proposing this pathway has mitohormetic effects, then actually demonstrate the metabolic benefits, not just weight loss, would be necessary.

We performed a GTT and ITT on the PDSS2^{BKO} animals in order to have a more thorough metabolic characterization. In the interest of time constraints for the revision, we did these GTT and ITT measurements on male mice fed a defined, low CoQ diet for 2 weeks and we did not see significant differences between groups (Fig. EV5G-J). For further insight into the glucose metabolism within the model we also measured gluconeogenic gene expression in the liver of mice fed a high fat diet. We observed that PEPCK and glucose-6-phosphatase have increased expression in PDSS2^{BKO} livers (Fig. 7J), which is in line with past studies that have shown that FGF21 increases hepatic gluconeogenesis. Further, to show the action of FGF21 secreted from BAT on target tissues we added data showing that

expression of Glut1, an FGF21 mediator (Ge, X. et al., 2011), is increased in extra-BAT tissue in PDSS2^{BKO} animals including eWAT, liver and muscle (Fig. 7I).

4. In Figure 2, the authors included results showing how mitochondrial fission and Drp1 phosphorylation at serine 616 (previously shown to be PKA dependent) are affected by CoQ2 inhibitor 4CBA. This information is not well integrated to the whole paper as of right now and seems out of place.

We show Drp1 phosphorylation status because in Fig. 2 we study mitochondrial morphology in the model we propose. In Fig. 2A we see that mitochondria in 4CBA treated cells have a more elongated phenotype compared to control cells, from this we hypothesized that fission of mitochondria may be impaired. Since Drp1 is an integral protein for mitochondrial fission we measured its phosphorylation/activation status, and it is in fact down which is in line with the observed phenotype. We have reworded the results section for Figure 2 to better incorporate this piece of data, but we feel as though measuring Drp1 phosphorylation gives insight into the morphological phenotype we see in Fig. 2A.

Minor points

1. There were some typos within the manuscript and sometimes it did make reading it a bit challenge. For example, ref #70 is basically an empty entry with only the first author partial name. Some additional proof reading would be helpful.

We have made sure to thoroughly reread and edit the manuscript for any typos before resubmitting.

Dear Dr Stahl,

Thank you for submitting your revised manuscript (EMBOJ-2023-114056R) to The EMBO Journal. As mentioned, your amended study was sent back to the referees for their re-evaluation, and we have received comments from two of them, which I enclose below. Please note that while referee #2 was unfortunately not able to re-evaluate the work at this time, we have assessed your response to the concerns raised editorially and found them to be addressed satisfactorily. As you will see, the other experts stated that the work has been substantially improved by the revisions and they are now broadly in favour of publication.

Thus, we are pleased to inform you that your manuscript has been accepted in principle for publication in The EMBO Journal.

We now need you to take care of a number of minor issues related to formatting and data presentation as detailed below, which should be addressed at re-submission.

Please contact me at any time if you have additional questions related to below points.

As you might have noted on our web page, every paper at the EMBO Journal now includes a 'Synopsis', displayed on the html and freely accessible to all readers. The synopsis includes a 'model' figure as well as 2-5 one-short-sentence bullet points that summarize the article. I would appreciate if you could provide this figure and the bullet points.

Thank you for giving us the chance to consider your manuscript for The EMBO Journal. I look forward to your final revision.

Again, please contact me at any time if you need any help or have further questions.

Best regards,

Daniel Klimmeck

>> Please add up to five keywords for your study.

>> Adjust the title of the 'Conflicts of Interests' section to 'Disclosure and Competing Interests Statement'.

>> Adjust the reference format to EMBO Journal style, limiting to 10 authors et al. .

>> EV figures: You can enter up to 5 EV Figures. The figure files need to be uploaded as individual, high-resolution figure files and labeled "Figure EV1" etc., and the legends need to be added to the manuscript, after the main figure legends. Any additional content should be merged with their legends into one 'Appendix' PDF and renamed "Appendix Figure S1" etc. The appendix will need a ToC with page numbers on the first page. The source data files and figure callouts in the manuscript will need to be adjusted accordingly.

>> The "Expanded View Table 1" in the manuscript should be renamed "Reagents and Tools table" and uploaded as a separate file.

>> Data availability section: please add a URL to the GSE165940 dataset. Adjust the statement to 'Other data that supports the findings of this manuscript can be found in this article or ...'.

>> Consider additional changes and comments from our production team as indicated below:

-Figure legends:

Please note that legend for figure 6H is incorrectly labelled as G.

Please indicate the statistical test used for data analysis in the legends of figures 1c, e-h; 2b, c, e; 3b-d; 4a-i; 5b-f; 6a-g, i, j; 7a-c, e-f, h-j; EV1a, d-f; EV2a-d, f, h; EV3a-d; EV4a-b; EV5b, c, e, h, j; EV6a-f, h, i; 7b-d, f-k.

Please note that in figures 2b-c, e; EV1f; EV5b, c, e, h, j there is a mismatch between the annotated p values in the figure legend and the annotated p values in the figure file that should be corrected.

Please define the annotated p values ### in the legend of figure 4i.

Please define the annotated p values ## in the legend of figure EV4b

Referee #1:

The authors have provided a significant amount of new experimental data, thereby addressing all initial major concerns raised by the reviewers.

Referee #3:

The authors experimentally addressed most of the concerns raised. I have no further comments.

The authors addressed the minor editorial issues.

Dear Dr Andreas Stahl,

Thank you for submitting the revised version of your manuscript. I have now evaluated your amended manuscript and concluded that the remaining minor concerns have been sufficiently addressed.

Thus, I am pleased to inform you that your manuscript has been accepted for publication in the EMBO Journal.

Please note that it is The EMBO Journal policy for the transcript of the editorial process (containing referee reports and your response letter) to be published as an online supplement to each paper. Related, I would accordingly like to ask for your consent on keeping the additional referee figures included in this file.

If you do NOT want the transparent process file published, you will need to inform the Editorial Office via email immediately. More information is available here: https://www.embopress.org/transparent-process#Review_Process

On a different note, I would like to alert you that EMBO Press offers a format for a video-synopsis of work published with us, which essentially is a short, author-generated film explaining the core findings in hand drawings, and, as we believe, can be very useful to increase visibility of the work. This has proven to offer a nice opportunity for exposure i.p. for the first author(s) of the study. Please see the following link for representative examples and their integration into the article web page:

<https://www.embopress.org/doi/full/10.15252/embj.2019103932>

Finally, we have noted that the submitted version of your article is also posted on the preprint platform bioRxiv. We would appreciate if you could alert bioRxiv on the acceptance of this manuscript at The EMBO Journal in order to allow for an update of the entry status. Thank you in advance!

If you have any questions, please do not hesitate to call or email the Editorial Office.

Best regards,

Daniel Klimmeck

Daniel Klimmeck, PhD
Senior Editor
The EMBO Journal
EMBO
Postfach 1022-40
Meyerhofstrasse 1

D-69117 Heidelberg
contact@embojournal.org
Submit at: <http://emboj.msubmit.net>

-